# CORRUPTING MULTIPLICATION-BASED UNLEARNABLE DATASETS WITH PIXEL-BASED IMAGE TRANSFORMATIONS

## ABSTRACT

*Unlearnable datasets* (UDs) lead to a drastic drop in the generalization performance of models trained on them by introducing elaborate and imperceptible perturbations into clean training sets. Many existing defenses, *e.g.*, JPEG compression and adversarial training, effectively counter UDs based on norm-constrained additive noise. However, a fire-new type of multiplication-based UDs have been proposed and render existing defenses all ineffective, presenting a greater challenge to defenders. To address this, we express the multiplication-based unlearnable sample as the result of multiplying a matrix by a clean sample in a simplified scenario, and formalize the *intra-class matrix inconsistency* as $\Theta_{imi}$, *inter-class matrix consistency* as $\Theta_{imc}$ to investigate the working mechanism of the multiplication-based UDs. We conjecture that increasing both of these metrics will mitigate the unlearnability effect. Through validation experiments that commendably support our hypothesis, we further design a random matrix to boost both $\Theta_{imi}$ and $\Theta_{imc}$, achieving a notable degree of defense effect. Additionally, we have also designed two new forms of multiplication-based UDs, and demonstrate that our defense is effective against both of these UDs as well. Hence, by building upon and extending these facts, *we first propose a brand-new image **CO**rruption that employs randomly multiplicative transformation via **IN**terpolation operation* (**COIN**) *to successfully defend against multiplication-based UDs.* Our approach leverages global pixel random interpolations, effectively suppressing the impact of multiplicative noise in multiplication-based UDs. Extensive experiments demonstrate that our defense approach outperforms state-of-the-art defenses against multiplication-based UDs, achieving an improvement of 19.17%-44.63% in average test accuracy on the CIFAR-10 and CIFAR-100 dataset.

## 1 INTRODUCTION

The triumph of *deep neural networks* (DNNs) hinges on copious high-quality training data, motivating many commercial enterprises to scrape images from unidentified sources. In this scenario, adversaries may introduce elaborate and imperceptible perturbations to each image in the dataset, thereby creating an *unlearnable dataset* (UD) that is subsequently disseminated online. This manipulation ultimately leads to a diminished generalization capacity of the victim model after being trained on such a dataset. Previous UDs were devoted to applying additive perturbations under the constraint of $\mathcal{L}_p$ norm to ensure their visual concealment, *i.e.*, bounded UDs (Huang et al., 2021; Fowl et al., 2021; Tao et al., 2021; Fu et al., 2022; Yu et al., 2022; Sandoval-Segura et al., 2022b; Ren et al., 2023; Chen et al., 2023; Wen et al., 2023; Wu et al., 2023).

Correspondingly, many defense schemes (Tao et al., 2021; Liu et al., 2023b; Qin et al., 2023b; Dolatabadi et al., 2023; Segura et al., 2023) against bounded UDs have been proposed. Among them, the most outstanding and widely-used defense solutions are techniques like *adversarial training* (AT) (Tao et al., 2021) and JPEG compression (Liu et al., 2023b), as demonstrated in Fig. 1 (a)[1]. The ease with which bounded UDs are successfully defended can be attributed to the fact that the introduced additive noise is limited, which renders the noise distribution easily disrupted.

---

[1]The accuracy results of these bounded UDs are reproduced based on their released source codes.

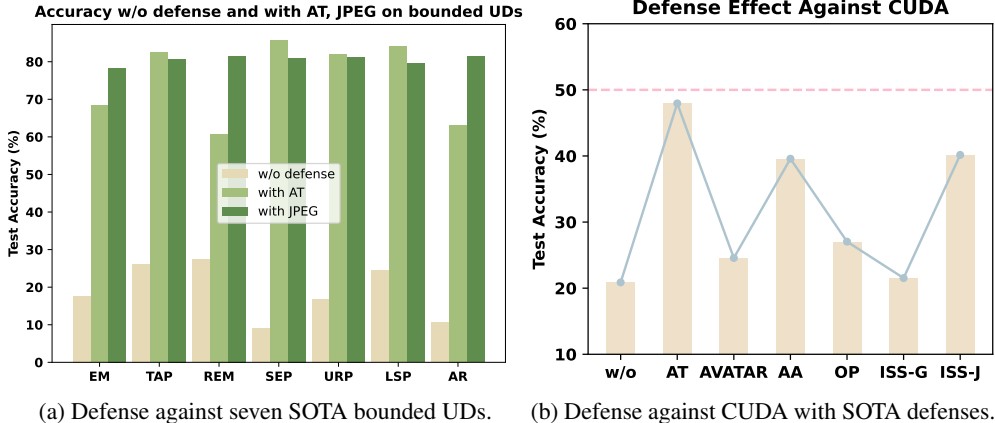

(a) Defense against seven SOTA bounded UDs.      (b) Defense against CUDA with SOTA defenses.

Figure 1: (a) Using AT and JPEG is enough to effectively defend against so many SOTA bounded UDs. (b) Using existing SOTA defense schemes is powerless against CUDA. The accuracy results of Fig. 1 (a) (using ResNet18), and (b) (using ResNet50) are obtained on the CIFAR-10 dataset using SGD with a momentum of 0.9, a learning rate of 0.1, and a batch size of 128 for training 80 epochs. The compression factor of JPEG is set to 10 as suggested by Liu et al. (2023b).

However, the latest proposed UD that employs multiplicative convolution operations without norm constraints (*i.e.*, multiplication-based UDs) (Sadasivan et al., 2023) has tremendously shaken the existing circumstances. Specifically, *convolution-based multiplication-based UD* (CUDA) (Sadasivan et al., 2023) has expanded the scope by using multiplication to broaden the noise spectrum, causing existing defense schemes to be completely ineffective as shown in Fig. 1 (b). Furthermore, as of now, no tailored defense against multiplication-based UDs has been proposed, presenting a significant and unprecedented challenge to defenders.

*To the best of our knowledge, none of the existing defense mechanisms demonstrate efficacy in effectively mitigating multiplication-based UDs.*

Given this context, there is an urgent need to formulate a defense paradigm against the prevailing multiplication-based UDs to tackle the challenges at hand. For designing a custom defense against multiplication-based UDs, we first align with Min et al. (2021); Javanmard & Soltanolkotabi (2022); Sadasivan et al. (2023) in simplifying the image samples as column vectors generated by a *Gaussian mixture model* (GMM) (Reynolds et al., 2009). In this manner, existing multiplication-based unlearnable samples can be expressed as the product of a matrix and clean samples. This multiplicative matrix can be understood as multiplicative perturbations, in contrast to the additive perturbations in most bounded UDs.

Meanwhile, Sadasivan et al. (2023) find that the test accuracy breathtakingly surpasses 90% when adding universal multiplicative noise to the dataset, which implies that it is not the multiplicative noise itself that renders the dataset unlearnable. Hence, the reasons behind the effectiveness of multiplication-based UDs remain to be further investigated. Inspired by the proposition from Yu et al. (2022) that the linearity separability property of noise is the reason for the effectiveness of UDs, we conjecture that either increasing the inconsistency within intra-class multiplicative noise or enhancing the similarity within inter-class multiplicative noise can both impair unlearnable effects. Back to the previously mentioned scenario with perfect multiplicative expression of unlearnable samples, we first formally define these two metrics as $\Theta_{imi}$ and $\Theta_{imc}$ customized for multiplication-based UDs. Then we conduct validation experiments based on these two quantitative indicators, consequently supporting our hypothesis. Now we are just motivated to design a transformation technique for the multiplicative matrix to effectively boost $\Theta_{imi}$ and $\Theta_{imc}$, and then extend this technique to defend against real unlearnable images. Specifically, we leverage a uniform distribution to generate random values and random shifts to construct a new random matrix, which is universally applied to left-multiply unlearnable samples. This transformation simultaneously achieves an enhancement in both $\Theta_{imi}$ and $\Theta_{imc}$, subsequently improving the test accuracy.

By expanding this random matrix to high-dimensional real multiplication-based unlearnable samples, *we first propose COIN, a newly designed defense strategy for countering multiplication-based*

*UDs, employing randomly multiplicative image transformation as its mechanism.* Concretely, we first sample random variables $\omega$ and $m$ from a uniform distribution. Afterwards, $m$ is used to obtain random pixels from unlearnable images for interpolation, and then we convert unlearnable samples to new samples via bilinear interpolation, involving $\omega$ for randomness. Extensive experiments reveal that our approach significantly overwhelms existing defense schemes, ranging from 19.17%-44.63% in test accuracy on CIFAR-10 and CIFAR-100. Our contributions are summarized as:

- We are the first to focus on defenses against multiplication-based UDs and the first to propose two brand-new metrics $\Theta_{imi}$ and $\Theta_{imc}$ tailored for multiplication-based UDs to explore the underlying mechanism of them.
- To the best of our knowledge, we propose the first highly effective defense strategy against multiplication-based UDs, termed as COIN, which utilizes a random pixel-based transformation and serves as a vital complement to the community of defense efforts against UDs.
- We further propose two new forms of multiplicative-based UDs in the context of GMM, and validate the effectiveness of our defense approach against them.
- Extensive experiments against existing multiplication-based UDs on three benchmark datasets and six commonly-used model architectures validate the effectiveness of our defense strategy.

## 2  RELATED WORK

### 2.1  UNLEARNABLE DATASETS

Current unlearnable datasets can be classified into two categories, *i.e.*, bounded UDs and multiplication-based UDs. The methods for crafting bounded UDs are as follows: Huang et al. (2021) first introduce the concept of "unlearnable examples" and utilize a dual minimization optimization approach to generate additive unlearnable noise with a restricted range. Subsequently, generation methods based on targeted adversarial samples (Fowl et al., 2021), universal random noise (Tao et al., 2021), robust unlearnable examples (Fu et al., 2022), linearly separable perturbations (Yu et al., 2022), autoregressive processes (Sandoval-Segura et al., 2022b), one-pixel short-cut (Wu et al., 2023)[2], and self-ensemble checkpoints (Chen et al., 2023) are successively proposed. Nonetheless, recent studies have indicated that popularly used defense techniques like AT and JPEG compression can readily counteract existing bounded UDs (Tao et al., 2021; Liu et al., 2023b).

Recently, a new type of multiplication-based UDs have been newly proposed, *i.e.*, Sadasivan et al. (2023) employ multiplicative convolutional operations to generate multiplicative noise without norm constraints. Unfortunately, all currently available defense methods prove ineffective against it, and there is currently no research exploring viable defense strategies.

### 2.2  DEFENSES AGAINST UDS

There are many defense techniques proposed for UDs so far. Tao et al. (2021) experimentally and theoretically demonstrate AT can effectively defend against bounded UDs, and Liu et al. (2023b) discover that simple image transformation techniques, *i.e.*, grayscale transformation (*a.k.a.*, ISS-G), and JPEG compression can also effectively defend against UDs. Thereafter, Qin et al. (2023b) employ *adversarial augmentations* (AA), Dolatabadi et al. (2023) purify UDs through diffusion models (AVATAR), while Segura et al. (2023) train a linear regression model to perform *orthogonal projection* (OP) on unlearnable samples. Nevertheless, none of these defense methods are tailor-made for multiplication-based UDs, with all of them failing against CUDA (Sadasivan et al., 2023). In contrast, our proposed defense is specifically designed for multiplication-based UDs, which effectively safe-

| Defense↓ Metric→ | Effective for CUDA | No need for external models |
|---|---|---|
| AT | ○ | ● |
| ISS-G | ○ | ● |
| ISS-J | ○ | ● |
| AA | ○ | ● |
| AVATAR | ○ | ○ |
| OP | ○ | ○ |
| **COIN (Ours)** | ● | ● |

Figure 2: The defense effectiveness against CUDA and dependence on external models are presented. "●" denotes fully satisfying the condition.

---

[2] We treat OPS as a case of bounded UD with $\mathcal{L}_0 = 1$ similar to (Liu et al., 2023b; Qin et al., 2023a).

guards against existing multiplication-based UD CUDA, offering a viable solution to the current vulnerability of security threats brought by multiplication-based UDs. A detailed comparison of defense schemes can be found in Fig. 2.

## 3 EXPLAINING THE MECHANISM OF MULTIPLICATION-BASED UDS

### 3.1 THREAT MODEL

The attacker creates a multiplication-based unlearnable dataset by multiplying the carefully crafted perturbation $\delta_i$ in some way to each image $x_i$ in the training set $\mathcal{D}_c$, thus causing the model $F$ with parameter $\theta$ trained on this dataset to generalize poorly to a clean test distribution $\mathcal{D}$ (Huang et al., 2021; Fowl et al., 2021; Yu et al., 2022; Sandoval-Segura et al., 2022b; Chen et al., 2023; Sadasivan et al., 2023; Wu et al., 2023). Formally, the attacker expects to work out the following bi-level objective:

$$\max_{\theta} \mathbb{E}_{(x,y)\sim\mathcal{D}} \left[ \mathcal{L}\left( F\left(x;\theta_p\right), y\right)\right] \tag{1}$$

$$s.t. \ \theta_p = \arg\min_{\theta} \sum_{(x_i,y_i)\in\mathcal{D}_c} \mathcal{L}\left( F\left(x_i \otimes \delta_i; \theta\right), y_i\right) \tag{2}$$

where $(x_i, y_i)$ represents the clean data from $\mathcal{D}_c$, $\mathcal{L}$ is a loss function, *e.g.*, cross-entropy loss, and $\otimes$ represents some kind of multiplicative operation, while ensuring the modifications to $x_i$ are not excessive for preserving the concealment of the sample.

As for defenders, in the absence of any knowledge of clean samples $x_i$, they aim to perform certain operations on UDs to achieve the opposite goal of Eq. (1).

### 3.2 CHALLENGES

Yu et al. (2022) reveal that the effectiveness of bounded UDs can be attributed to the linear separability of additive noise. However, Segura et al. (2023) discover that not all bounded UDs exhibit this property, providing a counterexample (Sandoval-Segura et al., 2022b). Consequently, there is still no clear consensus on how bounded UDs are effective. More importantly, there are significant differences in the form of the noise between multiplication-based UDs and bounded UDs, *e.g.*, class-wise multiplication-based perturbations from CUDA (Sadasivan et al., 2023) within the same class yet show non-identical noise, as illustrated in Fig. 9. This implies that the properties satisfied by the multiplicative noise corresponding to multiplication-based UDs may be different from those of bounded UDs. *Therefore, we are motivated to design custom evaluation metrics for assessing the "multiplication linearity separability" properties of multiplication-based UDs, aiming to investigate the reasons behind the effectiveness of this new type of UDs.*

### 3.3 PRELIMINARIES

Similar to Min et al. (2021); Javanmard & Soltanolkotabi (2022); Sadasivan et al. (2023), we define a binary classification problem involving a Bayesian classifier (Friedman et al., 1997), and the clean dataset $\mathcal{D}_c$ is sampled from a Gaussian mixture model $\mathcal{N}(y\boldsymbol{\mu}, I)$. Here, $y$ represents the labels $\{\pm 1\}$, with mean $\boldsymbol{\mu} \in \mathbb{R}^d$, and covariance $I \in \mathbb{R}^{d \times d}$ as the identity matrix ($d$ represents the feature dimension). We denote the clean sample as $x \in \mathbb{R}^d$, the multiplication-based unlearnable example $x_u$ in existing multiplication-based UDs can be formulated as left-multiplying the class-wise matrices $\mathcal{A}(a_y)$ by $x$, formulated as:

$$x_u = \mathcal{A}(a_y) \cdot x \tag{3}$$

where $a_y$ is a parameter used to create a multiplicative matrix $\mathcal{A}$ (*i.e.*, multiplicative noise) with respect to label $y$. Specifically, CUDA employs a tridiagonal matrix $\mathcal{A}_c(a_y)$, characterized by diagonal elements equal to 1, with the lower and upper diagonal elements set to $a_y$. We have further designed multiplicative UDs in the forms of *upper triangular matrices* and *lower triangular matrices* to demonstrate the generalizability of our proposed defense approach in Section 3.6. The more intuitive forms of these matrices are provided in the Appendix A.1.1.

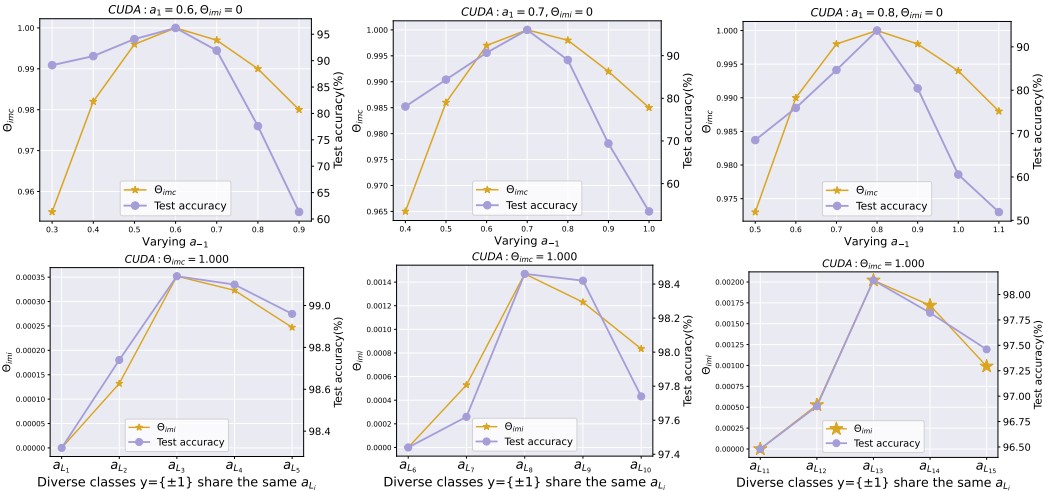

Figure 3: We explore the impact of $\Theta_{imc}$ (top row) and $\Theta_{imi}$ (bottom row) on test accuracy by manipulating the parameter $a_y$ in the process of generating CUDA datasets.

## 3.4 HYPOTHESIS AND VALIDATIONS

**Definition 1:** *We define the **intra-class matrix inconsistency**, denoted as $\Theta_{imi}$, as follows: Given the multiplicative matrices $\{\mathcal{A}_i \mid i = 1, 2, \cdots, n\}$ within a certain class $y_k$ (containing $n$ samples), we have an intra-class average matrix defined as $\mathcal{A}_{\mu_k} = \frac{1}{n}\sum_{i=1}^{n}\mathcal{A}_i$, an intra-class matrix variance defined as $\mathbb{D}_k = \frac{1}{n}\sum_{i=1}^{n}(\mathcal{A}_i - \mathcal{A}_{\mu_k})^2 \in \mathbb{R}^{d \times d}$, an intra-class matrix variance mean value defined as $\mathcal{V}_{mk} = \frac{1}{d^2}\sum_{i=0}^{d-1}\sum_{j=0}^{d-1}\mathbb{D}_k[i][j]$, and then we have $\Theta_{imi} = \frac{1}{c}\sum_{k=0}^{c-1}\mathcal{V}_{mk}$, where $c$ denotes the number of classes in $\mathcal{D}_c$.*

**Definition 2:** *We define the **inter-class matrix consistency**, denoted as $\Theta_{imc}$, as follows: Given the flattened intra-class average matrices of the $j$-th and $k$-th classes $flat(\mathcal{A}_{\mu_j})$, $flat(\mathcal{A}_{\mu_k})$, we then have $\Theta_{imc} = sim(flat(\mathcal{A}_{\mu_j}), flat(\mathcal{A}_{\mu_k}))$, where $flat()$ denotes flattening the matrix into a row vector and $sim(\cdot, \cdot)$ denotes cosine similarity, i.e., $sim(u, v) = uv^T/(\|u\|\|v\|)$.*

Inspired by the linear separability of additive noise in most bounded UDs, our intuition is that by decreasing the consistency of intra-class multiplicative matrices or increasing the consistency of inter-class multiplicative matrices, we can make the noise information introduced by the matrices more disordered and then less susceptible to be captured. This leads to classifiers failing to learn information unrelated to image features, thus resulting in an increase in test accuracy. In light of this, we propose our hypothesis as follows:

**Hypothesis 1:** *When $a_y$ in the multiplicative matrix is within a reasonable range, increasing $\Theta_{imi}$ or $\Theta_{imc}$ can both improve the test accuracy of classifiers trained on multiplication-based UDs, and vice versa.*

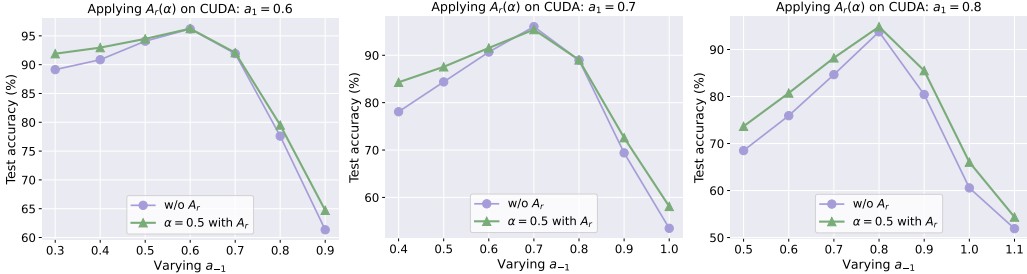

Figure 4: Comparison results of test accuracy before and after left-multiplying a random matrix $\mathcal{A}_r(\alpha)$ on all CUDA samples. $\alpha$ is all set to 0.5.

**Validation:** We first conduct experiments based on the preliminaries in Section 3.3 to construct CUDA datasets to validate our hypothesis. It can be observed from the top row of Fig. 3 that when

$\Theta_{imi}$ remains constant, an increase in $\Theta_{imc}$ corresponds to an improvement in test accuracy, whereas a decrease in $\Theta_{imc}$ leads to a decline in test accuracy, *i.e.*, $\Theta_{imc}$ and test accuracy exhibit the same trend of change. On the other hand, in the bottom row of Fig. 3, when $\Theta_{imc}$ is held constant, test accuracy increases as $\Theta_{imi}$ rises and decreases as $\Theta_{imi}$ falls. Hence, these experimental phenomena strongly support our proposed hypothesis. Additionally, we further explore in the top row Fig. 3, it is worth noting that when $\Theta_{imc}$=1.000 (*i.e.*, $a_1$ equals $a_{-1}$), it is equivalent to multiplying all clean samples by the same matrix $\mathcal{A}_c$, which obtains high test accuracy results of exceeding 90% regardless of $a_y$ equals 0.6, 0.7, or 0.8. The implementation details of these plots and the specific values of matrix lists $a_{L_1}$-$a_{L_{15}}$ are provided in Appendix A.3.

## 3.5 Our design: Random Matrix $\mathcal{A}_r$

Based on the hypothesis and supporting experimental results mentioned above, we are motivated to find a method that perturbs the distributions of multiplicative matrix $\mathcal{A}(a_y)$ in multiplication-based UDs for increasing $\Theta_{imi}$ and $\Theta_{imc}$, thereby improving test accuracy to achieve defense effect.

Our intuition is to further left-multiply $\mathcal{A}(a_y)$ by a random matrix $\mathcal{A}_r \in \mathbb{R}^{d \times d}$ to disrupt the matrix distribution. Concretely, to introduce randomness to the diagonal matrix $\mathcal{A}(a_y)$ for increasing $\Theta_{imi}$, we first set random values filled with the diagonal of $\mathcal{A}_r$. However, the form of $\mathcal{A}(a_y)$ remains unchanged by multiplying this $\mathcal{A}_r$, thus limiting the introduced randomness. Therefore, we add another set of random variables above the diagonal, but $\mathcal{A}_c$ still maintains the tridiagonal matrix structure. Thus, we shift variables by $m_i$ units simultaneously for each row to further enhance randomness. Due to the space limitation, more specific details regarding the above design reason on $\mathcal{A}_r$ are given in Appendix A.1.2. Next, we unify the random values and random shifts mentioned above using a uniform distribution $\mathcal{U}(-\alpha, \alpha)$, and ensure that the $\alpha$ is consistent for each class, thereby striving to enhance $\Theta_{imc}$ as much as possible while already improving $\Theta_{imi}$. During matrix $\mathcal{A}_r$ creation process, we first sample a variable $s \sim \mathcal{U}(-\alpha, \alpha, size = d), s \in \mathbb{R}^d$, and then obtain $m_i = \lfloor s_i \rfloor, n_i = s_i - m_i, 0 \leq i \leq d - 1$. The random matrix $\mathcal{A}_r$ is parameterized by $\alpha$ and is designed as:

$$\mathcal{A}_r(\alpha) = \begin{bmatrix} \underline{1 - n_0} & n_0 & 0 & 0 & \ldots & 0 \\ 0 & \underline{1 - n_1} & n_1 & 0 & \ldots & 0 \\ 0 & 0 & \underline{1 - n_2} & n_2 & \ldots & 0 \\ \vdots & \vdots & \vdots & \ddots & \vdots & \vdots \\ \vdots & \vdots & \vdots & \vdots & \ddots & \vdots \\ 0 & \ldots & \ldots & \ldots & 0 & \underline{1 - n_{d-1}} \end{bmatrix} \in \mathbb{R}^{d \times d} \tag{4}$$

where the $i$-th and $(i+1)$-th elements of the $i$-th row ($0 \leq i \leq d$-1) are $1 - n_i$ and $n_i$, and "$\underline{1 - n_i, n_i}$" means that the positions of these two elements for each row are shifted by $m_i$ units simultaneously. When the new location $i + m_i$ or $i + 1 + m_i$ exceeds the matrix boundaries, we take its modulus with respect to $d$ to obtain the new position. The implementation process of the $\mathcal{A}_r(\alpha)$ can be found in Algorithm 1.

Whereupon, we obtain test accuracy by left-multiplying all CUDA unlearnable samples with a random matrix $\mathcal{A}_r(\alpha)$, and then compare it with accuracy after training on the CUDA UD without $\mathcal{A}_r(\alpha)$. Firstly, both $\Theta_{imi}$ and $\Theta_{imc}$ increase indeed when $\mathcal{A}_r(\alpha)$ is employed as demonstrated in Tables 6 to 8. Secondly, we observe that the test accuracy with employing $\mathcal{A}_r$ is ahead of the accuracy obtained without using $\mathcal{A}_r$ regardless of $a_1$ is set to 0.6, 0.7, or 0.8 , as shown in Fig. 4, which is also consistent with our hypothesis. Ablation experiments on parameter $\alpha$ in $\mathcal{A}_r$ are in Appendix A.4.

## 3.6 Effectiveness of Our Defense Against Other Multiplication-based UDs

In order to further validate the effectiveness of our defense against multiplication-based UDs, we have newly designed two forms of multiplication-based UDs in GMM scenario, *i.e.*, we replace the $\mathcal{A}(a_y)$ in Eq. (3) with the matrices from Eq. (15) and Eq. (16), forming *upper triangular* and *lower triangular* multiplication-based UD respectively.

Consistent with the default settings in Fig. 4, after applying the defense using the self-designed random matrix $\mathcal{A}_r$, the test accuracy has been coherently improved, as shown in Tables 1 and 2.

| Defenses ↓ Values of $a_1, a_{-1} \rightarrow$ | 0.2, 0.8 | 0.2, 1.0 | 0.2, 1.2 | 0.2, 1.4 | 0.2, 1.6 |
|---|---|---|---|---|---|
| w/o | 92.2 | 80.6 | 64.6 | 55.2 | 52.0 |
| Applying $\mathcal{A}_r(0.5)$ | **95.2** | **91.0** | **84.4** | **74.2** | **64.2** |

Table 1: Test accuracy (%) results against upper triangle multiplication-based UD.

| Defenses ↓ Values of $a_1, a_{-1} \rightarrow$ | 0.2, 0.8 | 0.2, 1.0 | 0.2, 1.2 | 0.2, 1.4 | 0.2, 1.6 |
|---|---|---|---|---|---|
| w/o | 94.0 | 77.6 | 62.0 | 55.0 | 52.0 |
| Applying $\mathcal{A}_r(0.5)$ | **94.4** | **89.8** | **82.2** | **77.8** | **63.2** |

Table 2: Test accuracy (%) results against lower triangle multiplication-based UD.

*These results indicate that our designed defense solution is effective not only for tridiagonal matrix-based UDs from CUDA but also can be utilized to defend against attacks formed by other types of multiplicative matrices.*

## 4 METHODOLOGY

### 4.1 OUR DESIGN FOR COIN

Inspired by the linear interpolation process (Blu et al., 2004), we find that the two random values in each row of $\mathcal{A}_r$ can be effectively modeled as two weighting coefficients, and the random location offset $m_i$ in each row can be directly exploited to find the positions for interpolation. *Therefore, the previous process of left-multiplying the matrix $\mathcal{A}_r$ can be modeled as a random linear interpolation process to be applied for image transformations.* In view of this, we are able to directly extend random matrix $\mathcal{A}_r$ to be employed in real image domain. Unlike the previous sample $x \in \mathbb{R}^d$, the unlearnable image $x_u \in \mathbb{R}^{C \times H \times W}$ requires variables along with both horizontal and vertical directions and employing bilinear interpolation instead of linear interpolation. The formulaic definitions are as follows:

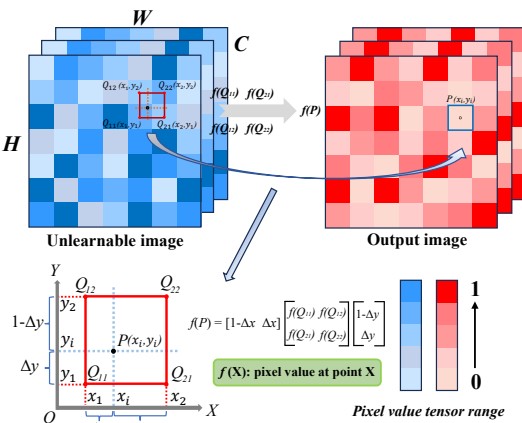

Figure 5: Our defense scheme COIN.

$$s_x, s_y \sim \mathcal{U}(-\alpha, \alpha, size = H \cdot W) \tag{5}$$

where $\mathcal{U}$ denotes a uniform distribution with size of height $H$ multiplied by width $W$, $\alpha$ controls the range of the generated random variables. Considering that $s_x$ and $s_y$ are both fundamentally arrays with size of $H \cdot W$, we obtain arrays $m_x$ and $m_y$ by rounding down each variable from the arrays to its integer part (*i.e.*, random location offsets), formulated as follows:

$$m_{xi} = \lfloor s_{xi} \rfloor, \quad m_{yi} = \lfloor s_{yi} \rfloor \tag{6}$$

where $\lfloor\ \rfloor$ represents the floor function, and $i$ is the index in the array, ranging from 0 to $H \cdot W - 1$ (with the same meaning in the equations below). Subsequently, the arrays with coefficients required for interpolation $\omega_x, \omega_y$ are computed as follows:

$$\omega_{xi} = s_{xi} - m_{xi}, \quad \omega_{yi} = s_{xi} - m_{yi} \tag{7}$$

To obtain the coordinates of the pixels used for interpolation, we first initialize a coordinate grid:

$$c_x, c_y = meshgrid(arange(W), arange(H)) \in \mathbb{R}^{H \times W} \tag{8}$$

where $meshgrid$ denotes coordinate grid creation function, $arange$ is employed to produce an array with values evenly distributed within a specified range. So now we can obtain the coordinates of the four nearest pixel points around the desired interpolation point in the bilinear interpolation process:

$$q_{11i} = ((c_{xi} + m_{xi})\%W, (c_{yi} + m_{yi})\%H) \tag{9}$$

$$q_{21_i} = ((c_{x_i} + m_{x_i} + 1)\%W, (c_{y_i} + m_{y_i})\%H) \tag{10}$$

$$q_{12_i} = ((c_{x_i} + m_{x_i})\%W, (c_{y_i} + m_{y_i} + 1)\%H) \tag{11}$$

$$q_{22_i} = ((c_{x_i} + m_{x_i} + 1)\%W, (c_{y_i} + m_{y_i} + 1)\%H) \tag{12}$$

where $\%$ represents the modulo function, ensuring that the horizontal coordinate ranges from 0 to $W - 1$ and the vertical coordinate ranges from 0 to $H - 1$. Hence, we obtain new pixel values by using the pixel values of the four points mentioned above through bilinear interpolation:

$$\mathcal{F}_j(p_i) = \begin{bmatrix} 1 - \omega_{x_i} & \omega_{x_i} \end{bmatrix} \begin{bmatrix} \mathcal{F}_j(q_{11_i}) & \mathcal{F}_j(q_{12_i}) \\ \mathcal{F}_j(q_{21_i}) & \mathcal{F}_j(q_{22_i}) \end{bmatrix} \begin{bmatrix} 1 - \omega_{y_i} \\ \omega_{y_i} \end{bmatrix} \tag{13}$$

where $\mathcal{F}_j()$ denotes the pixel value of the $j$-th channel at a certain coordinate point, $p_i$ denotes the coordinate of the newly generated pixel point. Each generated pixel value should be clipped within the range (0,1). Finally, we gain the transformed image $x_t$ by applying Eq. (13) and clipping to pixel values of each channel of the image $x_u$. The schematic diagram of the above process is shown in Fig. 5, and we summarize the complete process in Algorithm 2.

## 5 EXPERIMENTS

### 5.1 EXPERIMENTAL SETTINGS

Four widely-used network architectures including *ResNet* (RN) (He et al., 2016), *DenseNet* (DN) (Huang et al., 2017), *MobileNetV2* (MNV2) (Sandler et al., 2018), *GoogleNet* (GN) (Szegedy et al., 2015), *InceptionNetV3* (INV3) (Szegedy et al., 2016), and VGG (Simonyan & Zisserman, 2014) are selected. Meanwhile, three benchmark datasets CIFAR-10 (Krizhevsky & Hinton, 2009), CIFAR-100 (Krizhevsky & Hinton, 2009), and ImageNet100 (Deng et al., 2009)[3] are employed. The uniform distribution range $\alpha$ is set to 2.0. During training on existing multiplication-based UDs, we use SGD for training for 80 epochs with a momentum of 0.9, a learning rate of 0.1, and batch sizes of 128, 32 for CIFAR dataset, ImageNet100 dataset, respectively.

| Datasets | CIFAR-10 (Krizhevsky & Hinton, 2009) | | | | | | CIFAR-100 (Krizhevsky & Hinton, 2009) | | | | AVG |
|---|---|---|---|---|---|---|---|---|---|---|---|
| Defenses↓  Models→ | RN18 | VGG16 | DN121 | MNV2 | GN | INV3 | RN18 | VGG16 | DN121 | MNV2 | |
| w/o | 26.49 | 24.65 | 27.21 | 21.34 | 18.73 | 21.10 | 14.31 | 12.53 | 13.90 | 12.94 | 19.32 |
| MU (Zhang et al., 2018) | 26.72 | 28.07 | 24.67 | 24.63 | 26.04 | 24.99 | 17.09 | 13.35 | 19.97 | 13.55 | 21.91 |
| CM (Yun et al., 2019) | 26.02 | 28.53 | 24.64 | 20.73 | 20.11 | 24.25 | 12.51 | 10.14 | 20.77 | 10.14 | 19.78 |
| CO (DeVries & Taylor, 2017) | 20.07 | 27.58 | 24.86 | 20.46 | 18.87 | 26.06 | 12.80 | 10.56 | 16.19 | 13.56 | 19.10 |
| DP-SGD (Hong et al., 2020) | 25.50 | 23.02 | 25.25 | 25.78 | 17.65 | 21.18 | 12.42 | 10.56 | 16.36 | 12.72 | 19.04 |
| AT (Tao et al., 2021) | 50.59 | 45.95 | 49.01 | 42.59 | 50.62 | 47.66 | 37.27 | 28.18 | 34.21 | 35.74 | 42.18 |
| AVATAR (Dolatabadi et al., 2023) | 30.67 | 29.57 | 33.15 | 28.53 | 30.40 | 24.68 | 14.49 | 10.81 | 12.97 | 13.85 | 22.91 |
| AA (Qin et al., 2023b) | 39.85 | 38.68 | 38.92 | 41.06 | 38.58 | 39.01 | 24.83 | 1.00 | 27.89 | 20.49 | 31.03 |
| OP (Segura et al., 2023) | 29.77 | 30.33 | 33.82 | 28.86 | 26.52 | 23.94 | 20.17 | 14.59 | 15.55 | 23.02 | 24.66 |
| ISS-G (Liu et al., 2023b) | 25.77 | 21.42 | 26.73 | 19.85 | 15.41 | 22.63 | 8.80 | 6.40 | 11.48 | 8.71 | 16.72 |
| ISS-J (Liu et al., 2023b) | 45.10 | 40.26 | 39.79 | 41.46 | 38.49 | 41.49 | 33.62 | 26.92 | 28.94 | 31.23 | 36.73 |
| **COIN (Ours)** | **71.90** | **73.65** | **70.45** | **73.63** | **72.88** | **69.07** | **48.63** | **46.74** | **45.72** | **48.53** | **61.35** |

Table 3: Test accuracy (%) on CIFAR-10 and CIFAR-100 with defenses against CUDA.

### 5.2 DEFENSE COMPETITORS

We compare our defense COIN with SOTA defenses against UDs, *i.e.*, AT (Tao et al., 2021), ISS (Liu et al., 2023b) (including JPEG compression, *a.k.a.*, ISS-J and grayscale transformation, *a.k.a.*, ISS-G), AVATAR (Dolatabadi et al., 2023), OP (Segura et al., 2023), and AA (Qin et al., 2023b). We also apply four defense strategies proposed by Borgnia et al. (2021), *i.e.*, *differential privacy SGD* (DP-SGD) (Hong et al., 2020),

| Defenses↓  Models→ | RN18 | RN50 | DN121 | MNV2 | AVG |
|---|---|---|---|---|---|
| w/o | 25.74 | 26.66 | 21.70 | 16.30 | 22.60 |
| MU (Zhang et al., 2018) | 34.96 | 19.38 | 27.78 | 15.60 | 24.43 |
| CM (Yun et al., 2019) | 16.54 | 24.04 | 23.58 | 8.00 | 18.04 |
| CO (DeVries & Taylor, 2017) | 25.46 | 29.20 | 23.90 | 17.58 | 24.04 |
| AT (Tao et al., 2021) | 37.82 | 36.80 | 30.34 | 41.42 | 36.60 |
| ISS-G (Liu et al., 2023b) | 14.92 | 13.50 | 9.78 | 5.78 | 11.00 |
| ISS-J (Liu et al., 2023b) | 30.10 | 37.04 | 25.52 | 28.04 | 30.18 |
| **COIN (Ours)** | 37.80 | 35.38 | 35.22 | 41.50 | **37.48** |

Table 4: Test accuracy (%) on ImageNet with defenses against CUDA.

*cutmix* (CM) (Yun et al., 2019), *cutout* (CO) (DeVries & Taylor, 2017), and *mixup* (MU) (Zhang et al., 2018), which are popularly used to test whether it can defend against UDs. Consistent with previous works (Liu et al., 2023b; Tao et al., 2021; Qin et al., 2023b), we evaluate defense schemes with *test accuracy*, *i.e.*, the model accuracy on a clean test set after training on UDs.

### 5.3 EVALUATION ON OUR DEFENSE COIN

---

[3]We select the first 100 classes from the ImageNet dataset with image size of 224×224.

The test accuracy results for defense against existing multiplication-based UDs on benchmark datasets are presented in Tables 3 and 4. "**AVG**" denotes the average value for each row. The values covered by deep green denote the best defense effect, while values covered by light green denote the second-best defense effect.

It can be observed the average test accuracy (*i.e.*, values in **AVG** column) obtained by existing defense schemes against CUDA (Sadasivan et al., 2023) lag behind COIN by as much as 19.17%-44.63% as shown in Table 3, which demonstrates the superiority of our defense. Meanwhile, our defense scheme also maintains a leading advantage of 0.88%-26.48% on large datasets as shown in Table 4. The reason ISS-J largely lags behind COIN against CUDA can be deduced from Fig. 9, *i.e.*, the excessive global multiplicative noise introduced by CUDA results in a significant loss of features after the lossy compression from ISS-J. Additional explorations in COIN against bounded UDs are demonstrated in Appendix A.5.

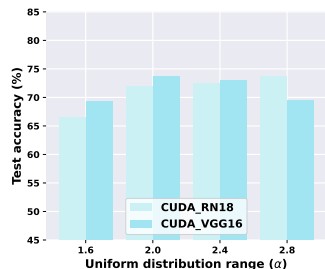

Figure 6: Using COIN against CUDA with varying $\alpha$ on RN18 and VGG16.

## 5.4 ABLATION EXPERIMENTS ON $\alpha$

We investigate the impact of uniform distribution range $\alpha$ on the effectiveness of our defense COIN, as shown in Fig. 6. The test accuracy against CUDA increases initially with the rise in $\alpha$ and then starts to decrease. This is because initially, as $\alpha$ increases, the CUDA perturbations gradually become disrupted. However, as $\alpha$ continues to increase, excessive corruptions damage image features, leading to a deterioration in defense effect. We opt for an $\alpha$ of 2.0 that yields the highest average defense effect.

## 5.5 TIME COMPLEXITY ANALYSIS FOR COIN

Assuming that the time complexity of each line of code in Algorithm 2 is $\mathcal{O}(1)$, then for an multiplication-based UD $\mathcal{D}_u = \{x_{ui} \in \mathbb{R}^{C \times H \times W}\}_{i=1}^{N}$, the overall time complexity of performing COIN is $\mathcal{O}(N \times C \times H \times W) + \mathcal{O}(N \times H \times W) + \mathcal{O}(N)$. Due to the fact that the values of $C$, $H$, and $W$ of image $x_{ui}$ are constant, *e.g.*, $C$=3, $H$=32, and $W$=32 for CIFAR-10 images, the final time complexity is optimized to: $\mathcal{O}(N) + \mathcal{O}(N) + \mathcal{O}(N) = \mathcal{O}(N)$. We then employ multiple defense strategies when training ResNet18 on CUDA CIFAR-10 for 80 epochs, and measure their corresponding time overheads, as shown in Fig. 7. We can conclude that our defense approach COIN is relatively efficient compared with existing defense schemes.

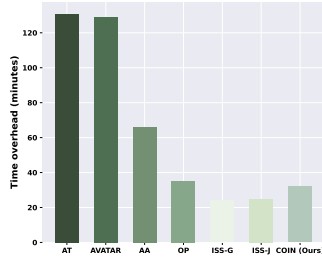

Figure 7: Time overhead on CIFAR-10 using RN18.

## 6 CONCLUSION

In this paper, we demonstrate for the first time that existing defense mechanisms against UDs are all ineffective against multiplication-based UDs. In light of this, we focus for the first time the challenging issue of defending against multiplication-based UDs. Subsequently, we model the process of existing multiplication-based UDs based on GMMs and Bayesian binary problems by applying multiplicative matrices to samples. Simultaneously, we discover that the consistency of intra-class and inter-class noise in multiplication-based UDs has a profound impact on the unlearnable effect, then we define two quantitative metrics $\Theta_{imi}$ and $\Theta_{imc}$ and investigate how to mitigate the impact of multiplicative matrices. We find increasing both of these two metrics can mitigate the effectiveness of multiplication-based UDs and design a new random matrix to increase both metrics. In addition, we have newly designed two different types of multiplication-based UDs and experimentally validated the effectiveness of our defense approach against all of them. Furthermore, in the context of real samples and based on the above ideas, we first propose a novel image transformation based on global pixel-level random resampling via bilinear interpolation, which universally guards against existing multiplication-based UDs. Extensive experiments on various benchmark datasets and widely-used models verified the effectiveness of our defense.

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

# A APPENDIX

## A.1 INTUITIVE DISPLAYS OF MATRICES

### A.1.1 INTUITIVE DISPLAYS OF $\mathcal{A}(a_y)$ AND $\mathcal{A}_r(\alpha)$

During CUDA UDs (Sadasivan et al., 2023) generation process, the tridiagonal matrix $\mathcal{A}_c(a_y)$ is designed as:

$$
\mathcal{A}_c(a_y) = \begin{bmatrix}
1 & a_y & 0 & 0 & \dots & 0 \\
a_y & 1 & a_y & 0 & \dots & 0 \\
0 & a_y & 1 & a_y & \dots & \vdots \\
\vdots & \vdots & \vdots & \ddots & \ddots & 0 \\
\vdots & \vdots & \vdots & \vdots & \ddots & a_y \\
0 & \dots & \dots & 0 & a_y & 1
\end{bmatrix} \in \mathbb{R}^{d \times d}
\tag{14}
$$

where constant $a_y$ corresponding to different $y$ is also different.

As for our newly designed upper triangular matrix, which is defined as:

$$
\mathcal{A}_u(a_y) = \begin{bmatrix}
1 & a_y & 0 & 0 & \dots & 0 \\
0 & 1 & a_y & 0 & \dots & 0 \\
0 & 0 & 1 & a_y & \dots & \vdots \\
\vdots & \vdots & \vdots & \ddots & \ddots & 0 \\
\vdots & \vdots & \vdots & \vdots & \ddots & a_y \\
0 & \dots & \dots & 0 & 0 & 1
\end{bmatrix} \in \mathbb{R}^{d \times d}
\tag{15}
$$

where constant $a_y$ corresponding to different $y$ is also different. Similarly, the lower triangular matrix is defined as:

$$
\mathcal{A}_l(a_y) = \begin{bmatrix}
1 & 0 & 0 & 0 & \dots & 0 \\
a_y & 1 & 0 & 0 & \dots & 0 \\
0 & a_y & 1 & 0 & \dots & \vdots \\
\vdots & \vdots & \vdots & \ddots & \ddots & 0 \\
\vdots & \vdots & \vdots & \vdots & \ddots & 0 \\
0 & \dots & \dots & 0 & a_y & 1
\end{bmatrix} \in \mathbb{R}^{d \times d}
\tag{16}
$$

where constant $a_y$ corresponding to different $y$ is also different.

---

**Algorithm 1:** Matrix $\mathcal{A}_r(\alpha)$ creation process

---

    **Input:** Feature dimension $d$; matrix strength $\alpha$.
    **Output:** Matrix $\mathcal{A}_r(\alpha)$.
    **Function:** Uniform distribution $\mathcal{U}$.
**1** Randomly sample $s = \mathcal{U}(-\alpha, \alpha, size = d)$;
**2** Round down to the integer $m = \lfloor s \rfloor$;
**3** Fractional part $n = s - m$;
**4** Initialize a matrix with all zeros $\mathcal{A}_r(\alpha) \in \mathbb{R}^{d \times d}$;
**5** **for** $i$ = 0 to $(d-1)$ **do**
**6**      **for** $j$ = 0 to $(d-1)$ **do**
**7**          **if** $i + m[i] < 0$ **then**
**8**              $x = i + m[i] + d - 1$;
**9**          **end**
**10**          **else if** $i + m[i] > d - 1$ **then**
**11**              $x = i + m[i] - d + 1$;
**12**          **end**
**13**          **else**
**14**              $x = i + m[i]$;
**15**          **end**
**16**          **if** $x == j$ **then**
**17**              $\mathcal{A}_r(\alpha)[i, j] = 1 - n[i]$;
**18**          **end**
**19**          **else if** $(x + 1)\%d == j$ **then**
**20**              $\mathcal{A}_r(\alpha)[i, j] = n[i]$;
**21**          **end**
**22**      **end**
**23** **end**
**24** **Return:** Matrix $\mathcal{A}_r(\alpha)$.

---

During matrix $\mathcal{A}_r$ creation process, we first sample a variable $s \sim \mathcal{U}(-\alpha, \alpha, size = d), s \in \mathbb{R}^d$, and then obtain $m_i = \lfloor s_i \rfloor, n_i = s_i - m_i, 0 \leq i \leq d - 1$. The random matrix $\mathcal{A}_r$ is parameterized by $\alpha$ and is designed as:

$$\mathcal{A}_r(\alpha) = \begin{bmatrix} \underline{1 - n_0} & n_0 & 0 & 0 & \dots & 0 \\ 0 & \underline{1 - n_1} & n_1 & 0 & \dots & 0 \\ 0 & 0 & \underline{1 - n_2} & n_2 & \dots & 0 \\ \vdots & \vdots & \vdots & \ddots & \vdots & \vdots \\ \vdots & \vdots & \vdots & \vdots & \ddots & \vdots \\ 0 & \dots & \dots & \dots & 0 & \underline{1 - n_{d-1}} \end{bmatrix} \in \mathbb{R}^{d \times d} \quad (17)$$

where the $i$-th and $(i+1)$-th elements of the $i$-th row ($0 \leq i \leq d$-1) are $1 - n_i$ and $n_i$, and "$\underline{1 - n_i}, n_i$" means that the positions of these two elements for each row are shifted by $m_i$ units simultaneously. When the new location $i + m_i$ or $i + 1 + m_i$ exceeds the matrix boundaries, we take its modulus with respect to $d$ to obtain the new position. The detailed process can be referred to in Algorithm 1.

### A.1.2 WHY DO WE DESIGN $\mathcal{A}_r$ LIKE THIS?

We will explain the rationale behind the design of the $\mathcal{A}_r$ matrix. First of all, we define a diagonal matrix $\mathcal{A}_o$, which might be a new matrix representation of a potential multiplication-based attack similar to bounded UD OPS (Wu et al., 2023), and a matrix $\mathcal{A}_{tr}$ with random values on its diagonal

elements formulated as:

$$\mathcal{A}_o\left(a_y, i\right) = \begin{bmatrix} 1 & 0 & 0 & 0 & \ldots & 0 \\ 0 & \ddots & 0 & 0 & \ldots & 0 \\ 0 & 0 & 1 & 0 & \ldots & 0 \\ \vdots & \vdots & \vdots & a_y & \vdots & \vdots \\ \vdots & \vdots & \vdots & \vdots & 1 & \vdots \\ 0 & \ldots & \ldots & 0 & 0 & 1 \end{bmatrix} \in \mathbb{R}^{d \times d} \tag{18}$$

$$\mathcal{A}_{tr} = \begin{bmatrix} n_0 & 0 & 0 & 0 & \ldots & 0 \\ 0 & n_1 & 0 & 0 & \ldots & 0 \\ 0 & 0 & \ddots & 0 & \ldots & 0 \\ \vdots & \vdots & \vdots & n_i & \vdots & \vdots \\ \vdots & \vdots & \vdots & \vdots & \ddots & \vdots \\ 0 & \ldots & \ldots & 0 & 0 & n_{d-1} \end{bmatrix} \in \mathbb{R}^{d \times d} \tag{19}$$

where $n_0, n_1, ..., n_{d-1}$ are all random values. If we multiply matrix $\mathcal{A}_{tr}$ by $\mathcal{A}(a_y)$, we will get new matrices as:

$$\mathcal{A}_{tr} \cdot \mathcal{A}_c(a_y) = \begin{bmatrix} n_0 & n_0 a_y & 0 & 0 & \ldots & 0 \\ n_1 a_y & n_1 & n_1 a_y & 0 & \ldots & 0 \\ 0 & \ddots & \ddots & \ddots & \ldots & 0 \\ \vdots & \vdots & n_i a_y & n_i & n_i a_y & \vdots \\ \vdots & \vdots & \vdots & \ddots & \ddots & \vdots \\ 0 & \ldots & \ldots & 0 & n_{d-1}a_y & n_{d-1} \end{bmatrix} \in \mathbb{R}^{d \times d} \tag{20}$$

$$\mathcal{A}_{tr} \cdot \mathcal{A}_o(a_y) = \begin{bmatrix} n_0 & 0 & 0 & 0 & \ldots & 0 \\ 0 & n_1 & 0 & 0 & \ldots & 0 \\ 0 & 0 & \ddots & 0 & \ldots & 0 \\ \vdots & \vdots & \vdots & n_i a_y & \vdots & \vdots \\ \vdots & \vdots & \vdots & \vdots & \ddots & \vdots \\ 0 & \ldots & \ldots & 0 & 0 & n_{d-1} \end{bmatrix} \in \mathbb{R}^{d \times d} \tag{21}$$

Apparently, setting random variables only on the diagonal like $\mathcal{A}_{tr}$ does not alter the original form of the multiplicative matrix, *i.e.*, $\mathcal{A}_c$ remains a tridiagonal matrix, and $\mathcal{A}_o$ remains a diagonal matrix, which has a very limited effect on disrupting the distribution of the original matrix.

So, how would adding another set of random variables above the diagonal change the form of the matrix? Thus, we define a new matrix $\mathcal{A}_{sr}$ as:

$$\mathcal{A}_{sr} = \begin{bmatrix} 1 - n_0 & n_0 & 0 & 0 & \ldots & 0 \\ 0 & 1 - n_1 & n_1 & 0 & \ldots & 0 \\ 0 & 0 & 1 - n_2 & n_2 & \ldots & 0 \\ \vdots & \vdots & \vdots & \ddots & \vdots & \vdots \\ \vdots & \vdots & \vdots & \vdots & \ddots & \vdots \\ 0 & \ldots & \ldots & \ldots & 0 & 1 - n_{d-1} \end{bmatrix} \in \mathbb{R}^{d \times d} \tag{22}$$

where $n_0, n_1, ..., n_{d-1}$ are all random values. Similarly, we multiply it with $\mathcal{A}(a_y)$ to obtain the following result:

$$\mathcal{A}_{sr} \cdot \mathcal{A}_c\left(a_y\right) = \begin{bmatrix} 1 - n_0 + n_0 a_y & (1 - n_0)a_y + n_0 & n_0 a_y & 0 & \ldots & 0 \\ (1 - n_1)a_y & 1 - n_1 + n_1 a_y & (1 - n_1)a_y + n_1 & n_1 a_y & \ldots & 0 \\ 0 & (1 - n_2)a_y & 1 - n_2 + n_2 a_y & (1 - n_2)a_y + n_2 & \ldots & \vdots \\ \vdots & \vdots & \vdots & \ddots & \ddots & 0 \\ \vdots & \vdots & \vdots & \vdots & \ddots & n_{d-1}a_y \\ 0 & \ldots & \ldots & 0 & (1 - n_{d-1})a_y & (1 - n_{d-1}) + n_{d-1}a_y \end{bmatrix} \tag{23}$$

$$\mathcal{A}_{sr} \cdot \mathcal{A}o(a_y, i) = \begin{bmatrix} 1 - n_0 & n_0 & 0 & \dots & \dots & 0 \\ 0 & 1 - n_1 & n_1 & 0 & \dots & 0 \\ \vdots & \vdots & \ddots & \ddots & \dots & \vdots \\ \vdots & \vdots & \vdots & (1 - n_i)a_y & n_i & \dots \\ \vdots & \vdots & \vdots & \vdots & \ddots & \ddots \\ 0 & \dots & \dots & \dots & \dots & 1 - n_{d-1} \end{bmatrix} \tag{24}$$

It can be observed that the matrix $\mathcal{A}_o$ has completely altered its form, accommodating more randomness. However, $\mathcal{A}_c$ still maintains the tridiagonal matrix form. So, in order to alter the form of $\mathcal{A}_c$ as much as possible for introducing as much randomness as possible, it seems reasonable to add another set of random values below the diagonal of $\mathcal{A}_{sr}$. However, this idea encounters two challenging practical issues: **I:** *Adding more diagonal values would significantly increase the computational cost of matrix multiplication, which is not conducive to efficient algorithms.* **II:** *More diagonal values correspond to more noise, which will harm sample features.* Therefore, this way is not feasible. So, we come up with the idea of introducing small random offsets to the random variables for each row. This approach both does not introduce new variables also breaks the original tridiagonal form of the matrix $\mathcal{A}_c(a_y)$, thus further increasing randomness.

### A.2 FURTHER STUDY ON IMAGE CORRUPTIONS

The defense strategy we propose, along with ISS-G and ISS-J, fundamentally fall within the domain of corruption techniques (Liu et al., 2023a; Hu et al., 2023). Hence, we aim to explore additional image corruption operations to ascertain the possibility of more potential defenses against multiplication-based UDs, as illustrated in Fig. 8. It can be observed that the majority of these commonly used image corruptions are largely ineffective in defending against multiplication-based UDs. Delving deeper into the defense against these image corruption techniques for multiplication-based UDs would be a meaningful avenue for future research.

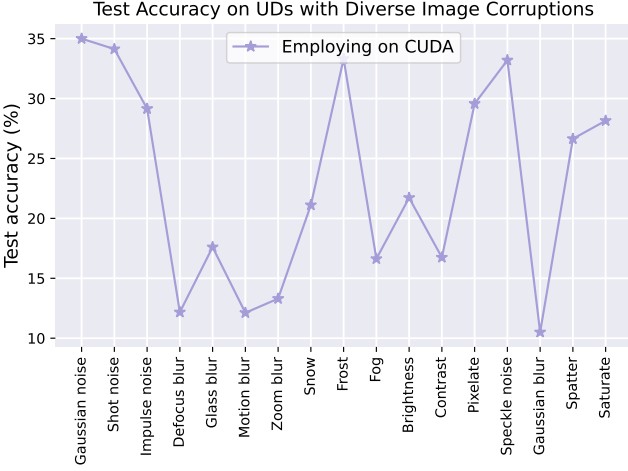

Figure 8: Test accuracy (%) on CIFAR-10 using RN18 with corruptions against CUDA.

| Matrix list $a_{L_i}$ | $a_{L_1}$ | $a_{L_2}$ | $a_{L_3}$ | $a_{L_4}$ | $a_{L_5}$ |
|---|---|---|---|---|---|
| $a_y$ values in list | [0.4] | [0.2, 0.3, 0.4] | [0.03, 0.06, 0.1, 0.2, 0.3, 0.4] | [0.05, 0.1, 0.2, 0.3, 0.4] | [0.1, 0.2, 0.3, 0.4] |
| Matrix list $a_{L_i}$ | $a_{L_6}$ | $a_{L_7}$ | $a_{L_8}$ | $a_{L_9}$ | $a_{L_{10}}$ |
| $a_y$ values in list | [0.5] | [0.3, 0.5, 0.7] | [0.05, 0.1, 0.5, 0.7] | [0.1, 0.5, 0.7] | [0.2, 0.5, 0.7] |
| Matrix list $a_{L_i}$ | $a_{L_{11}}$ | $a_{L_{12}}$ | $a_{L_{13}}$ | $a_{L_{14}}$ | $a_{L_{15}}$ |
| $a_y$ values in list | [0.6] | [0.4, 0.6, 0.8] | [0.05, 0.1, 0.3, 0.6, 0.9] | [0.1, 0.2, 0.4, 0.6, 0.8, 0.9] | [0.2, 0.4, 0.6, 0.8] |

Table 5: The specific values of the matrix lists $a_{L1}$-$a_{L15}$ are shown. Each $a_y$ value corresponds to the matrix $\mathcal{A}_c(a_y)$, and the matrix lists $a_{Li}$ used for both classes are identical.

| w/o $\mathcal{A}_r$, $a_{-1} \rightarrow$ | 0.3 | 0.4 | 0.5 | 0.6 | 0.7 | 0.8 | 0.9 |
|---|---|---|---|---|---|---|---|
| $\Theta_{imi}$ | 0 | 0 | 0 | 0 | 0 | 0 | 0 |
| $\Theta_{imc}$ | 0.955 | 0.982 | 0.996 | 1.000 | 0.997 | 0.990 | 0.980 |
| $\mathcal{A}_r(0.5)$, $a_{-1} \rightarrow$ | 0.3 | 0.4 | 0.5 | 0.6 | 0.7 | 0.8 | 0.9 |
| $\Theta_{imi}$ | 0.000664 | 0.000663 | 0.000674 | 0.000698 | 0.000738 | 0.000782 | 0.000845 |
| $\Theta_{imc}$ | 0.981 | 0.993 | 0.998 | 1.000 | 0.999 | 0.996 | 0.992 |

Table 6: The specific values of $\Theta_{imi}$ and $\Theta_{imc}$ before and after using $\mathcal{A}_r$ on CUDA samples with $a_1 = 0.6$ and varying $a_{-1}$ in Fig. 4.

### A.3 IMPLEMENTATION DETAILS OF VALIDATION EXPERIMENTS

Similar to CUDA (Sadasivan et al., 2023), we sample 1000 samples from the Gaussian Mixture Model (with $\|\boldsymbol{\mu}\|_2$=2.0, $d$=100) as described in Section 3.3, and evenly split them into a training set (used for generating new training sets by CUDA) and a test set. We train a Naive Bayes classifier on the new training set and then calculate the test accuracy on the test set with the trained classifier. The results of test accuracy reported in Section 3.4 are the average results obtained after running the experiments 10 times with diverse random seeds.

| w/o $\mathcal{A}_r$, $a_{-1} \rightarrow$ | 0.4 | 0.5 | 0.6 | 0.7 | 0.8 | 0.9 | 1.0 |
|---|---|---|---|---|---|---|---|
| $\Theta_{imi}$ | 0 | 0 | 0 | 0 | 0 | 0 | 0 |
| $\Theta_{imc}$ | 0.965 | 0.986 | 0.997 | 1.000 | 0.998 | 0.992 | 0.985 |
| $\mathcal{A}_r(0.5)$, $a_{-1} \rightarrow$ | 0.4 | 0.5 | 0.6 | 0.7 | 0.8 | 0.9 | 1.0 |
| $\Theta_{imi}$ | 0.000701 | 0.000712 | 0.000734 | 0.000770 | 0.000822 | 0.000877 | 0.000953 |
| $\Theta_{imc}$ | 0.986 | 0.994 | 0.999 | 1.000 | 0.999 | 0.997 | 0.994 |

Table 7: The specific values of $\Theta_{imi}$ and $\Theta_{imc}$ before and after using $\mathcal{A}_r$ on CUDA samples with $a_1 = 0.7$ and varying $a_{-1}$ in Fig. 4.

| w/o $\mathcal{A}_r$, $a_{-1} \rightarrow$ | 0.5 | 0.6 | 0.7 | 0.8 | 0.9 | 1.0 | 1.1 |
|---|---|---|---|---|---|---|---|
| $\Theta_{imi}$ | 0 | 0 | 0 | 0 | 0 | 0 | 0 |
| $\Theta_{imc}$ | 0.973 | 0.990 | 0.998 | 1.000 | 0.998 | 0.994 | 0.988 |
| $\mathcal{A}_r(0.5)$, $a_{-1} \rightarrow$ | 0.5 | 0.6 | 0.7 | 0.8 | 0.9 | 1.0 | 1.1 |
| $\Theta_{imi}$ | 0.000762 | 0.000785 | 0.000818 | 0.000865 | 0.000929 | 0.000996 | 0.001083 |
| $\Theta_{imc}$ | 0.989 | 0.996 | 0.999 | 1.000 | 0.999 | 0.997 | 0.995 |

Table 8: The specific values of $\Theta_{imi}$ and $\Theta_{imc}$ before and after using $\mathcal{A}_r$ on CUDA samples with $a_1 = 0.8$ and varying $a_{-1}$ in Fig. 4.

In the process of constructing the CUDA UD, $\mathcal{A}_c(a_1)$ represents the left-multiplying matrix for samples of class 1, and $\mathcal{A}_c(a_{-1})$ is the left-multiplying matrix for samples of class -1.

In the top three subplots of Fig. 4, we apply class-wise matrices separately to all samples of class 1 or class -1, *i.e.*, all samples within the same class receive the same matrix to ensure $\Theta_{imi}$=0. Then, by changing the value of $a_{-1}$ (while keeping $a_1$ fixed), we can achieve variations in $\Theta_{imc}$.

In the bottom three subplots of Fig. 4, to investigate the impact of $\Theta_{imi}$ on test accuracy, we first designed a list of matrices, denoted as $a_{L_i}$, which contains several matrices. For each sample in a specific class, we randomly selected a matrix from the list to perform the matrix multiplication. By controlling the diversity of matrices in the list, we could control the variations in $\Theta_{imi}$. We set the matrix lists for each class to be the same to make $\Theta_{imc}$ remain fixed at 1.000. The specific configuration of the matrix list $a_{L_i}$ is shown in the table below:

It can be observed that from $a_{L_1}$ to $a_{L_5}$, $a_{L_6}$ to $a_{L_{10}}$, and $a_{L_{11}}$ to $a_{L_{15}}$, the fluctuations in the matrix lists increase, leading to larger values of $\Theta_{imi}$ accordingly.

### A.4 ADDITIONAL VALIDATION EXPERIMENTAL DETAILS AND RESULTS

#### A.4.1 ABLATION VALIDATION EXPERIMENTS ON $\alpha$ OF $\mathcal{A}_r(\alpha)$

Building upon Fig. 4, we further explore the impact of different $\alpha$ parameters of $\mathcal{A}_r(\alpha)$ on the final defense effectiveness. The values of $\Theta_{imi}$, $\Theta_{imc}$, and test accuracy results after applying $\mathcal{A}_r(\alpha)$

| $\mathcal{A}_r(0.4), a_1 = 0.6, a_{-1} \rightarrow$ | 0.3 | 0.4 | 0.5 | 0.6 | 0.7 | 0.8 | 0.9 |
|---|---|---|---|---|---|---|---|
| $\Theta_{imi}$ | 0.000446 | 0.000446 | 0.000454 | 0.000470 | 0.000497 | 0.000527 | 0.000569 |
| $\Theta_{imc}$ | 0.976 | 0.991 | 0.998 | 1.000 | 0.998 | 0.995 | 0.990 |
| Test accuracy (%) | 92.08 | 93.10 | 94.70 | 96.44 | 92.08 | 79.18 | 64.18 |
| $\mathcal{A}_r(0.4), a_1 = 0.7, a_{-1} \rightarrow$ | 0.4 | 0.5 | 0.6 | 0.7 | 0.8 | 0.9 | 1.0 |
| $\Theta_{imi}$ | 0.000471 | 0.000478 | 0.000495 | 0.000522 | 0.000551 | 0.000593 | 0.000642 |
| $\Theta_{imc}$ | 0.982 | 0.993 | 0.998 | 1.000 | 0.999 | 0.996 | 0.992 |
| Test accuracy (%) | 84.74 | 86.22 | 92.68 | 95.60 | 88.24 | 71.98 | 56.94 |
| $\mathcal{A}_r(0.4), a_1 = 0.8, a_{-1} \rightarrow$ | 0.5 | 0.6 | 0.7 | 0.8 | 0.9 | 1.0 | 1.1 |
| $\Theta_{imi}$ | 0.000513 | 0.000529 | 0.000552 | 0.000583 | 0.000625 | 0.000670 | 0.000728 |
| $\Theta_{imc}$ | 0.987 | 0.995 | 0.999 | 1.000 | 0.999 | 0.997 | 0.994 |
| Test accuracy (%) | 73.18 | 80.30 | 88.08 | 94.66 | 85.14 | 65.32 | 54.00 |

Table 9: The values of $\Theta_{imi}$, $\Theta_{imc}$, and test accuracy after applying $\mathcal{A}_r(0.4)$ on CUDA samples with $a_1 = 0.6, 0.7, 0.8$ and corresponding varying $a_{-1}$.

| $\mathcal{A}_r(0.6), a_1 = 0.6, a_{-1} \rightarrow$ | 0.3 | 0.4 | 0.5 | 0.6 | 0.7 | 0.8 | 0.9 |
|---|---|---|---|---|---|---|---|
| $\Theta_{imi}$ | 0.000931 | 0.000929 | 0.000943 | 0.000976 | 0.001032 | 0.001093 | 0.001183 |
| $\Theta_{imc}$ | 0.985 | 0.994 | 0.999 | 1.000 | 0.999 | 0.997 | 0.994 |
| Test accuracy (%) | 91.82 | 92.78 | 94.10 | 96.08 | 92.10 | 79.48 | 64.96 |
| $\mathcal{A}_r(0.6), a_1 = 0.7, a_{-1} \rightarrow$ | 0.4 | 0.5 | 0.6 | 0.7 | 0.8 | 0.9 | 1.0 |
| $\Theta_{imi}$ | 0.000982 | 0.000997 | 0.001027 | 0.001077 | 0.001149 | 0.001227 | 0.001334 |
| $\Theta_{imc}$ | 0.989 | 0.996 | 0.999 | 1.000 | 0.999 | 0.997 | 0.995 |
| Test accuracy (%) | 84.08 | 87.32 | 91.40 | 95.44 | 89.06 | 72.80 | 58.32 |
| $\mathcal{A}_r(0.6), a_1 = 0.8, a_{-1} \rightarrow$ | 0.5 | 0.6 | 0.7 | 0.8 | 0.9 | 1.0 | 1.1 |
| $\Theta_{imi}$ | 0.001067 | 0.001098 | 0.001145 | 0.001210 | 0.001300 | 0.001394 | 0.001518 |
| $\Theta_{imc}$ | 0.992 | 0.997 | 0.999 | 1.000 | 0.999 | 0.998 | 0.996 |
| Test accuracy (%) | 73.62 | 80.80 | 88.16 | 94.90 | 85.62 | 66.56 | 54.62 |

Table 10: The values of $\Theta_{imi}$, $\Theta_{imc}$, and test accuracy after applying $\mathcal{A}_r(0.6)$ on CUDA samples with $a_1 = 0.6, 0.7, 0.8$ and corresponding varying $a_{-1}$.

with varying $\alpha$ on CUDA are demonstrated in Tables 9 and 10. We compare these test accuracy results with original accuracy results (*i.e.*, w/o applying $\mathcal{A}_r$) in Tables 6 to 8. It can be found that there is an improvement in test accuracy when $\alpha$ is set to 0.4 or 0.6, indicating the presence of the defense effect. Furthermore, after applying $\mathcal{A}_r$, both $\Theta_{imi}$ and $\Theta_{imc}$ values increase, which also aligns with the conclusions drawn from Fig. 4.

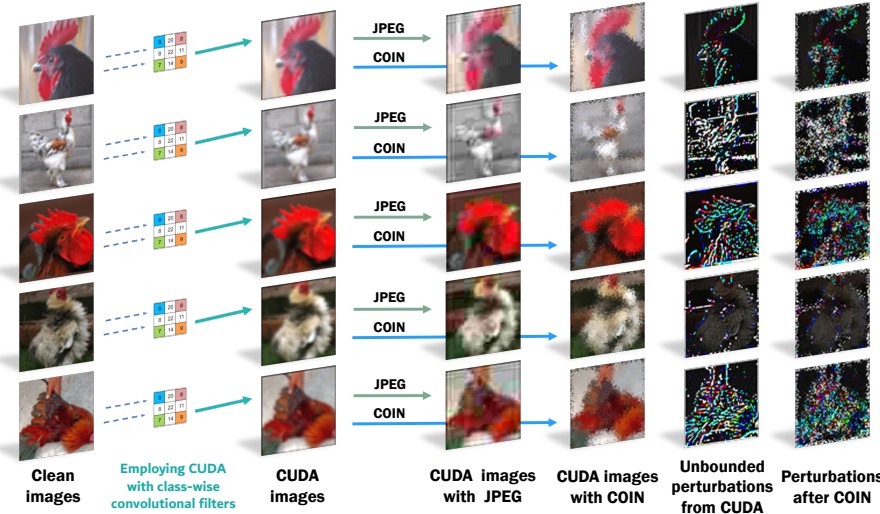

Figure 9: Clean images, CUDA images, CUDA images with JPEG defense and our defense COIN, and perturbations from class of 'cock' of the ImageNet UD.

## A.5 Additional main experimental results and details

### A.5.1 Additional main experiments on bounded UDs

While our COIN defense scheme is custom-designed for existing multiplication-based UDs, we are also interested in its effectiveness against bounded UDs. We select four SOTA unlearnable techniques for generating bounded UDs (Fowl et al., 2021; Tao et al., 2021; Chen et al., 2023; Wu et al., 2023) and compare the results in test accuracy before and after using COIN, as shown in Table 12. Specific experimental parameter settings are completely consistent with the main experiments. The details of reproducing these three bounded UDs are given in Appendix A.5.2.

From the results in this table, we can see that COIN has effective defense capabilities against URP, TAP, and OPS. However, its defense effectiveness against SEP is limited and falls below an acceptable level. Therefore, while COIN greatly excels in defending against multiplication-based UDs, it still has limitations when it comes to defending against bounded UDs.

### A.5.2 Additional details

Standard data augmentations like random cropping and random flipping are adopted in our experiments. As for the generation process of multiplication-based UDs, we also utilize their open-source code as follows:

- During generating CUDA UDs (Sadasivan et al., 2023), we directly run their official codes with default parameters `https://github.com/vinusankars/Convolution-based-Unlearnability`.

Remarks: Due to insufficient GPU capacity, a batch size of 64 was set when applying AA on CIFAR-10, CIFAR-100 using ResNet50.

During exploring COIN on bounded UDs, we mostly utilize their official codes. Detailed processes are as follows:

- For generating the TAP UD (Fowl et al., 2021), we simply crafted targeted adversarial exmples through PGD method (Madry et al., 2018) with iteration of 40, step size of 2/255 and perturbation budget of 8/255 in $\ell_\infty$-norm based on their unlearnable scheme.
- For generating the URP UD (Huang et al., 2021; Tao et al., 2021), we run the official codes `https://github.com/TLMichael/Delusive-Adversary`.
- For generating the SEP UD (Chen et al., 2023), we run their official codes `https://github.com/Sizhe-Chen/SEP` to produce UDs using ResNet18 checkpoints.
- During generating the OPS UD (Wu et al., 2023), we also run their official codes with default parametes `https://github.com/cychomatica/One-Pixel-Shotcut`.

## A.6 Gaining a Visual Insight into Defense Methods

We visually demonstrate *unlearnable images crafted by CUDA* (CUDA images), along with their clean images and perturbations as shown in Fig. 9. We obtain "class-wise multiplication-based perturbations from CUDA" by subtracting the corresponding clean images from the CUDA images. The perturbations from CUDA within the same class yet show non-identical noise, which differs to the class-wise form we understand in additive noise, and such noise does not exhibit linear separability like previous bounded UDs. To prove this, we train a linear logistic regression model on CUDA perturbations and report train accuracy following OP (Segura et al., 2023) and their official code, as shown in Table 11. This indicates that the added perturbations in CUDA are different from class-wise bounded perturbations, and indeed not totally linearly separable. Additionally, it can be observed that the JPEG transformation subjected to lossy compression tends to lose more features after employing CUDA images, which may be one of the reasons why JPEG compression fails to effectively defend against CUDA. After applying our COIN defense to CUDA images, visually, we are able to distinguish the specific categories of the images.

| UDs | Train acc (%) | Is it linearly separable for added perturbations? |
|---|---|---|
| ∘EM (Huang et al., 2021) | 100 | YES |
| ∘Regions-4 (Sandoval-Segura et al., 2022a) | 100 | YES |
| ∘Random Noise | 100 | YES |
| CUDA (Sadasivan et al., 2023) | 77.12 | **NO** |

Table 11: The linear separability of perturbations from UDs ("∘" denotes class-wise noise patterns).

| Bounded UDs→ | URP (Tao et al., 2021) | | | TAP (Fowl et al., 2021) | | | SEP (Chen et al., 2023) | | | OPS (Wu et al., 2023) | | |
|---|---|---|---|---|---|---|---|---|---|---|---|---|
| Defense↓  Model→ | RN18 | VGG19 | **AVG** | RN18 | VGG19 | **AVG** | RN18 | VGG19 | **AVG** | RN18 | VGG19 | **AVG** |
| w/o | 16.80 | 16.53 | 16.66 | 26.16 | 27.81 | 26.98 | 9.01 | 12.70 | 10.86 | 28.39 | 20.15 | 24.27 |
| **COIN** | 81.11 | 77.85 | 79.48 | 76.41 | 71.26 | 73.84 | 48.09 | 46.72 | 47.41 | 80.13 | 74.58 | 77.36 |

Table 12: The test accuracy results on three CIFAR-10 bounded UDs with and w/o applying COIN.

---

**Algorithm 2:** Our defense COIN

---

**Input:** Unbounded unlearnable dataset $\mathcal{D}_u = \{(x_{ui}, y_i)|i = 1, 2, \cdots, N, x_{ui} \in \mathbb{R}^{C \times H \times W}\}$; range of random variables $\alpha$.

**Output:** Transformed dataset $\mathcal{D}_t = \{(x_{ti}, y_i)|i = 1, 2, \cdots, N, x_{ti} \in \mathbb{R}^{C \times H \times W}\}$.

**Function:** Coordinate grid function *meshgrid*; uniform distribution $\mathcal{U}$; *arange* function returns evenly spaced values within a given interval; *clip* function clips values outside the interval to the interval edges.

1 **for** $cnt = 1$ *to* $N$ **do**
2     Sample random horizontal and vertical variables $s_x, s_y \sim \mathcal{U}(-\alpha, \alpha, size = H \cdot W)$;
3     Initialize a coordinate grid $c_x, c_y = meshgrid(arange(W), arange(H))$;
4     Initialize a pixel value list $L_p = [\ ]$;
5     **for** $i = 0$ *to* $H \cdot W - 1$ **do**
6        Get a random horizontal and vertical location offset $m_{xi}$ and $m_{yi}$;
7        Calculate horizontal and vertical weight coefficient $\omega_{xi}$ and $\omega_{yi}$;
8        Obtain the coordinates of the nearest four points: $q_{11i}, q_{21i}, q_{12i}, q_{22i}$;
9        **for** $j = 1$ *to* $C$ **do**
10           Apply bilinear interpolation to obtain new pixel value $\mathcal{F}_j(p_i)$;
11           $\mathcal{F}_j(p_i) = clip(\mathcal{F}_j(p_i), 0, 1)$;
12        **end**
13        $L_p.append([\mathcal{F}_1(p_i), \mathcal{F}_2(p_i), \cdots, \mathcal{F}_C(p_i)])$;
14     **end**
15     Output transformed image $x_{ti}$ by filling the pixel values using $L_p$;
16 **end**
17 **Return:** Transformed dataset $\mathcal{D}_t = \{(x_{ti}, y_i)|i = 1, 2, \cdots, N, x_{ti} \in \mathbb{R}^{C \times H \times W}\}$.

---

