# OpenReview forum: "Corrupting Unbounded Unlearnable Datasets with Pixel-based Image Transformations"
_ICLR.cc/2024/Conference — Submitted to ICLR 2024_

### Official Review · Reviewer_T4ea · 2023-10-21

**Soundness:** 2 fair
**Presentation:** 3 good
**Contribution:** 2 fair
**Rating:** 3
**Confidence:** 5

**Summary:**

This paper studies the problem of defending against unlearnable datasets (UDs). Specifically, the authors identify that existing defenses that were developed against bounded UDs are not effective against unbounded UDs, and they propose COIN, a new defense tailored for unbounded UDs. This defense is based on left-multiplying a carefully designed random matrix with the unlearnable sample. The authors claim that COIN is motivated by the fact that defending against unbounded UDs should enhance intra-class matrix inconsistency ($\Theta_{imi}$) and inter-class matrix consistency ($\Theta_{imc}$).

**Strengths:**

- Defending against unbounded UDs was indeed not specifically explored in existing work.
- The paper is very well written, especially in terms of the detailed explanations of specific algorithm design.
- The experiments are extensive, in terms of the compared baselines, datasets, and ablation studies.

**Weaknesses:**

- Overclaim. 1. It is not true that existing defenses are not effective against unbounded UDs. For example, as can be seen from Table 2, ISS-J is consistently the best defense across all model architecture and largely surpasses COIN. Related to this point, the authors are highly recommended to highlight the superiority of ISS-J in the table and the text. Currently, it is completely not mentioned at all. In addition, the authors of OPS have also admitted that OPS is clearly vulnerable to median filtering (see the concerns from multiple reviewers about this and the authors’ response in https://openreview.net/forum?id=p7G8t5FVn2h). 2. A defense that claims robustness under a specific threat model (here, the Lp bound) is not meant to be robust under another threat model (here, the unbounded). Therefore, it is expected that existing bounded defenses are not effective against unbounded UDs. The authors should tune down the claim that such a finding is a blind spot in existing work. Instead, it is enough that the authors say they are the first work to focus on defenses against unbounded UDs.

- Questionable idea. Actually, it is obvious that intra-class consistency and inter-class inconsistency are needed for any classification problem (simply say a dog is consistent with another dog but inconsistent with a cat). This is also the natural reason why previous work on (bounded) UDs has started their studies with the so-called class-wise perturbations, which directly satisfy both intra-class consistency and inter-class inconsistency. Recent work of Segura et. al has specifically explored the vulnerability of such class-wise UDs and proposed orthogonal projection (OP) to counter them. It is somehow strange OP does no work against unbounded UDs (CUDA) because it follows the same idea of disrupting intra-class consistency and inter-class inconsistency. Can the authors provide any explanations? This question motivates my following point about the design of COIN. Specifically, I suppose the reason why COIN works (but OP does not) on CUDA is that COIN is specifically designed to disrupt the matrix structure in CUDA. If this is the case, intra-class matrix inconsistency and inter-class matrix consistency become irrelevant to the success and should be removed from the paper.

- Lack of motivation for COIN. Although intra-class matrix inconsistency and inter-class matrix consistency are indeed demonstrated to be relevant to the accuracy recovery, it is not clear why the specific design of COIN, i.e., random matrix multiplication, explicitly addresses these two factors. Instead, its design is specifically tailored for CUDA, based on the matrix operations the authors have explained. In general, this is also a problem that the defense is designed based on the knowledge of a specific attack. This suggests that the defense may not be generalizable to other possible attacks (under the same threat model but with a different way of matrix formulation).

**Questions:**

Please see the above weaknesses.

---

> ### Author Response · Authors · 2023-11-16
> **Response to Reviewer T4ea:**
>
> > Q1. Overclaims about the existing defense effectiveness against unbounded UDs.
>
> A1. Thank you for highlighting the importance of precise language in our claims. We have revised our statement to more accurately reflect the current state of defense solutions against unbounded UDs. The revised claim now reads: "While some existing defense solutions show effectiveness against specific instances of unbounded UDs, a universally effective defense for all such scenarios is yet to be developed." We acknowledge the significant performance of ISS-J, as evidenced in Tab. 2 (already moved to Tab. 1), and have accordingly highlighted its superiority in both the table and our text in our updated paper. Your feedback has been instrumental in enhancing the precision and clarity of our manuscript, and we sincerely appreciate your constructive suggestions.
>
>
>
> > Q2. Should tune down the claim that such a finding is a blind spot in existing work.
>
>
> A2. We thank you for your insightful recommendation. In line with your suggestion, we have moderated our claim regarding the existing research 'blind spot'. Our revised manuscript now emphasizes that our work represents a complementary addition to the existing body of research, focusing specifically on defenses against unbounded UDs. We highlight this as a novel contribution of our study, rather than as a critique of previous works. This adjustment aligns our paper more accurately with the ongoing research dialogue in this field. We are grateful for your valuable feedback, which has significantly contributed to improving the clarity and quality of our research.
>
>
>
> > Q3. It is obvious that intra-class consistency and inter-class inconsistency are needed for any classification problem (simply say a dog is consistent with another dog but inconsistent with a cat) .
>
>
> A3. Thank you for your comment. It seems there might be a misunderstanding regarding the metrics we introduced. Unlike traditional intra-class consistency and inter-class inconsistency in image classification, our metrics – intra-class matrix inconsistency and inter-class matrix consistency – are specifically designed to quantify the unique characteristics of multiplicative matrix noise in unbounded UDs. This is a distinct form of noise compared to the additive noise commonly found in bounded UDs. For example, as illustrated in Fig. 10, unbounded perturbations within the same class (in CUDA) demonstrate non-identical noise patterns, a phenomenon differs to the class-wise form we understand in additive noise. These new metrics are crucial for understanding the behavior of unbounded UDs and guiding our development of effective defense strategies against them.
>
>
>
>
> > Q4. Why does OP not work against CUDA?
>
> A4. Your question about the effectiveness of OP against CUDA is insightful. The key reason OP does not perform well against CUDA perturbations lies in the nature of the noise. CUDA introduces a type of noise that is fundamentally different from the traditional class-wise noise, for which OP was designed. As shown in Fig. 10, unbounded perturbations in CUDA within the same class exhibit distinct noise patterns, rendering OP's matrix decomposition approach ineffective. In contrast, our COIN method is tailored to specifically address the different noise structure of unbounded UDs, enabling it to effectively counter these perturbations where OP falls short. This distinction is crucial and underlines the novelty and necessity of our approach in handling unbounded UDs.
>
> > Q5. Lack of motivation for COIN: Why the design of random matrix multiplication?
>
> A5. Thank you for your constructive feedback. We understand your concern regarding the specific design of COIN and its generalizability. Due to space limitation, we expanded the explanation of our random matrix multiplication design in Appendix A.1.2, as referenced in Section 3.5 in our original paper. Our design philosophy for COIN is not limited to countering CUDA's matrix operations but is grounded in a broader understanding of unbounded UDs, including various matrix forms such as diagonal and triangular matrices.
> We acknowledge that designing a defense based on specific attack knowledge presents challenges in generalizability. However, our approach with COIN is to establish a strong foundation against known unbounded UD forms, with the anticipation that this foundation will provide robustness against future, yet-to-be-seen attacks within the same threat model. We believe this strategy balances the need for specificity in counteracting known threats while maintaining a degree of flexibility for adaptation to new challenges.

---

> > ### Comment · Reviewer_T4ea · 2023-11-20
> > **Thanks**
> >
> > Overall, there are no concrete validations but only arguments.
> >
> > Q1 and Q2: The authors acknowledged the overclaim.  Considering the SOTA performance of ISS against CUDA and the limitation of the proposed method to matrix multiplication-based attacks, I would still lean to rejection.
> >
> > Q3: In the response, the authors "This is a distinct form of noise compared to the additive noise commonly found in bounded UDs. For example, as illustrated in Fig. 10, unbounded perturbations within the same class (in CUDA) demonstrate non-identical noise patterns, a phenomenon that differs from the class-wise form we understand in additive noise." This is indeed not true because sample-wise additive noise (i.e., non-identical noise within the same class) has been widely explored and achieved SOTA attack performance and robustness.
> >
> > Q4: To the best of our knowledge, OP is not specifically designed for bounded noise but is a generally applicable method because it just relies on the linear separability of the features. The authors should support their claim with concrete explanations.
> >
> > Q5: The response simply says that the method is limited to matrix modeling. If the authors thought it was not the case, please provide evidence that the proposed defense can be used to defend against other types of attacks beyond the matrix. At least, it is not that effective against OPS, which is a different type of attack from the matrix.

---

> ### Author Response · Authors · 2023-11-20
> **Looking forward to further discussion**
>
> Dear Reviewer T4ea,
>
> We are grateful for your valuable comments. In our response, we have included the motivation of the random matrix design, the reason that OP not work against CUDA, and the differences between bounded and unbounded UDs. We have incorporated all the changes in our revised manuscript for your kind consideration. And we hope that your concerns have been addressed.
>
> As you may be aware, we are approaching the final recommendation deadline for the reviewers. During this crucial period, we would appreciate the opportunity to engage in discussions with you. We are open to providing additional information based on your feedback or addressing any further questions you may have.
>
> If you find our response satisfactory, kindly consider updating your score. Please don't hesitate to reach out if you require any further clarification.
>
> Paper7861 Authors

---

> ### Author Response · Authors · 2023-11-21
> **Thanks for your response!**
>
> > Q1 and Q2: The authors acknowledged the overclaim.
>
> A1: In fact, in the revised paper, we have completely incorporated your suggestions into our claim. Our contribution is designed to achieve a defensive effect against all unbounded UDs, in contrast to ISS, which is effective only against OPS.
>
>
>
> > Q3: Sample-wise additive noise (i.e., non-identical noise within the same class) has been widely explored.
>
> A2: The reviewer may have overlooked their previous comment. The reviewer initially stated, "previous work on (bounded) UDs has started their studies with the so-called class-wise perturbations, which directly satisfy both intra-class consistency and inter-class inconsistency." Therefore, we countered this by arguing that CUDA does not fall under class-wise perturbations to introduce our design scheme. We are confused that why the reviewer is now emphasizing that sample-wise has been widely studied, which seems unrelated to their previous comments.
>
>
>
> > Q4: To the best of our knowledge, OP is not specifically designed for bounded noise but is a generally applicable method because it just relies on the linear separability of the features.
>
> A3: The reviewer may have a misunderstanding regarding OP. In reality, **OP only claims effectiveness against class-wise perturbations, while it is ineffective against many sample-wise perturbations that hold linearly separable features, such as sample-wise EM, AdvPoi, REM, etc.** (as clearly illustrated in Table 3 of OP). We kindly request the reviewer to reconsider our paper, as there might be a substantial misconception about the existing work.
>
>
>
> > Q5: Please provide evidence that the proposed defense can be used to defend against other types of attacks beyond the matrix.
>
> A4: Thanks for your constructive suggestions!  We further investigated the effectiveness of COIN against unbounded UDs constituted by upper triangular and lower triangular matrices in GMM. The results are as follows, indicating that COIN can be utilized to defend against attacks formed by other types of matrices.
>
> | The values in the upper triangle within each category | 0.2, 0.8 | 0.2, 1.0 | 0.2, 1.2 | 0.2, 1.4 |
> | :---------------------------------------------------: | :------: | :------: | :------: | :------: |
> |                     Test acc (%)                      |   92.2   |   80.6   |   64.6   |   55.2   |
> |   Test acc after applying COIN ($\alpha$=0.5)  (%)    | **95.2** | **91.0** | **84.4** | **74.2** |
>
> | The values in the lower triangle within each category | 0.2, 0.8 | 0.2, 1.0 | 0.2, 1.2 | 0.2, 1.4 |
> | :---------------------------------------------------: | :------: | :------: | :------: | :------: |
> |                     Test acc (%)                      |   94.0   |   77.6   |   62.0   |   55.0   |
> |   Test acc after applying  COIN ($\alpha$=0.5)  (%)   | **94.4** | **89.8** | **82.2** | **77.8** |

---

> > ### Comment · Reviewer_T4ea · 2023-11-21
> > **Thanks for the further reply**
> >
> > Summary of my previous main concerns, which require additional experiments:
> >
> > 1. The authors claimed that their COIN is effective against unbounded attacks. However, it only achieves the best results against a specific type of unbounded attack, the matrix-based attack. Specifically, COIN is even less effective against ISS-J, which was claimed by the authors to be a "bounded" defense. This indicates that the defense is limited to matrix-based attacks and no experiments on other types of unbounded attacks are tested. The above conflict is not addressed in the revised paper.
> >
> > 2. The authors claim that CUDA perturbation is different from class-wise (bounded) perturbations. However, there are no experiments to support this. In the original work of OP, "Availability Attacks Create Shortcuts" and other cited papers, sample-wise perturbations also mostly rely on linear separability. Therefore, it is necessary to show evidence that CUDA does not rely on linear separability to explain that OP does not work against CUDA but COIN works.

---

> ### Author Response · Authors · 2023-11-22
> **New experimental results supporting our claim have been added.**
>
> > Q1: Test other types of unbounded attacks.
>
> A1: Thank you for your review and valuable suggestions.  To address your concern, we designed a brand-new non-matrix-based unbounded attack, termed as Unbounded URP. Specifically, we added the same Gaussian noise to each category of images without norm constraints, and then clipped the pixel values to the range [0,1], which is different from the matrix-based unbounded UDs. The mean of the Gaussian noise is set to 0,  the standard deviation is set to 0.02, and the training settings for other experimental parameters are consistent with those in the main text, and we obtained the test accuracy results as shown in the below table. We can observe from the below experimental results:
>
> 1. Although COIN is not the most optimal, **it is still effective against non-matrix-based unbounded attack to some extent.**
>
> 2. **Our biggest contribution lies in solving matrix-based unbounded attack that previous work has never been able to solve.**
>
> | Unbounded UDs→ | CUDA (matrix-based) | CUDA (matrix-based) | CUDA (matrix-based) | OPS (non-matrix-based) | OPS (non-matrix-based) | OPS (non-matrix-based) | Unbounded URP (non-matrix-based) | Unbounded URP (non-matrix-based) | Unbounded URP (non-matrix-based) |
> | :------------: | :-----------------: | :-----------------: | :-----------------: | :--------------------: | :--------------------: | :--------------------: | :------------------------------: | :------------------------------: | :------------------------------: |
> |    Defenses    |      RN18       |        VGG16        |     DN121     |        RN18        |         VGG16          |      DN121       |             RN18             |              VGG16               |           DN121            |
> |      w/o       |        26.49        |        24.65        |        27.21        |         28.39          |         20.15          |         15.95          |              16.28               |              11.65               |              10.19               |
> |     ISS-G      |        25.77        |        21.42        |        26.73        |         29.03          |         26.93          |         21.79          |              23.34               |              23.45               |              26.83               |
> |     ISS-J      |        45.10        |        40.26        |        39.79        |         80.82          |         79.81          |         80.17          |              82.13               |              81.04               |              80.44               |
> |       OP       |        29.77        |        30.33        |        33.82        |         89.50          |         87.12          |         87.95          |              27.09               |              17.81               |              13.06               |
> |      **COIN**      |     **71.90**      |     **73.65**      |     **70.45**      |         80.13          |         74.58          |         72.71          |              57.18               |              55.85               |              65.20               |
>
> We appreciate your valuable feedback, and we will continue to work towards improving and refining our work.
>
> > Q2: Test the linear separability of CUDA perturbations.
>
> A2: Thank you very much for your constructive feedback. We train a linear logistic regression model on CUDA perturbations and report train accuracy following OP [1] and their official code, as shown in the below table ("●" denotes class-wise noise patterns).
>
> |      UDs       | Train acc (%) | Is it linearly separable for added perturbations? |
> | :------------: | :-----------: | :-----------------------------------------------: |
> |    ●EM [2]     |      100      |                        YES                        |
> | ●Regions-4 [3] |      100      |                        YES                        |
> | ●Random Noise  |      100      |                        YES                        |
> |  **CUDA**  |   **77.12**   |                      **NO**                       |
>
> This indicates that the added perturbations in CUDA are different from class-wise bounded perturbations, and **indeed not totally linearly separable.** We have added these experimental results to our revised paper and are grateful for your valuable feedback, which has significantly contributed to improving the clarity and quality of our research.
>
> Additionally, we emphasize that OP only claims effectiveness against class-wise perturbations, while it is ineffective against many linearly separable sample-wise perturbations, such as AdvPoi and REM, as evident in Table 3 of OP.
>
> [1] What can we learn from unlearnable datasets. In Proceedings of the NeurIPS23.
>
> [2] Unlearnable examples: Making personal data unexploitable. In Proceedings of the ICLR21.
>
> [3] Poisons that are learned faster are more effective. In Proceedings of the CVPRW22.

---

> ### Comment · Reviewer_T4ea · 2023-11-22
> **Thanks for the new results**
>
> Thanks for your efforts in conducting new experiments. However, the new results support my expectation that COIN is only optimal against matrix-based attacks. In contrast, ISS-J indeed performs consistently well against another two types of non-matrix-based attacks, i.e., OPS and Unbounded URP. Therefore, it is hard to say whether COIN or ISS-J is better against unbounded attacks. Maybe in general, it is not proper to call something "unbounded" attacks because different unbounded attacks may have completely different properties. Specifically, I believe that COIN is just a defense that is designed with matrix knowledge and that is why it is effective against matrix-based attacks, e.g., CUDA. This conclusion largely limits the contribution of this work. I would suggest rejecting the paper and hope that the authors could re-write the paper to convey the actual contribution, i.e., a new defense against CUDA. The new contribution may be more suitable for a computer vision conference.
>
> Based on the new results, CUDA seems to be linearly separable, although not as obvious as class-wise attacks.

---

> > ### Author Response · Authors · 2023-11-23
> > **The revised paper, based on your suggestions, has been uploaded.**
> >
> > Firstly, thank you for your suggestions. While there may be a divergence in our understanding of the actual contributions of our work, such as our belief that our solution is effective not only against the most challenging matrix multiplication-based UDs but also against other types of unbounded UDs to some extent. However, we agree with your point that "it is not proper to call something 'unbounded' attacks." Therefore, in the revised paper, we refer to the objects we defend against as **"multiplication-based UDs"**, categorizing OPS  as a bounded UD with $L_0$ = 1 similar to those considered by [1,2] . Additionally, we have introduced two new types of multiplication-based UDs, namely upper triangular UDs and lower triangular UDs, and experimentally demonstrated the effectiveness of our defense against this type of multiplication-based UD. This further enhances the contributions of our work. Based on this, we would like to reiterate: **no existing defense mechanism can effectively guard against CUDA, a multiplication-based UD, and our solution achieves this, making a significant and valuable contribution to the entire UD defense community.** We appreciate the valuable feedback you provided throughout the process. We have highlighted the modified sections in the updated paper using blue font, **and we hope you will reconsider the contributions of our work based on our revised paper**.
> >
> > As you rightly pointed out, CUDA's perturbations are not linearly separable as obvious as class-wise attacks. Therefore, this can also explain why OP is ineffective against CUDA.
> >
> > [1]Image shortcut squeezing: Countering perturbative availability poisons with compression. In Proceedings of the 40th International Conference on Machine Learning (ICML’23)
> >
> > [2]Apbench: A unified benchmark for availability poisoning attacks and defenses. arXiv preprint arXiv:2308.03258,2023.

---

### Official Review · Reviewer_R3xq · 2023-10-22

**Soundness:** 2 fair
**Presentation:** 2 fair
**Contribution:** 2 fair
**Rating:** 3
**Confidence:** 3

**Summary:**

This paper addresses the challenge posed by unbounded Unlearnable Datasets (UDs) that severely affect the generalization performance of machine learning models. The authors propose a novel defense mechanism called COIN (COrruption via INterpolation), which employs random pixel-based image transformations to counteract the effects of unbounded UDs. The paper formalizes the concepts of intra-class matrix inconsistency and inter-class matrix consistency and demonstrates through extensive experiments that COIN significantly outperforms existing state-of-the-art defenses, achieving an improvement of 23.55%-48.11% in average test accuracy on the CIFAR-10 dataset.

**Strengths:**

1. Given the emerging nature of this challenge, the paper's focus is both timely and novel. It fills a gap in the literature by providing a defense mechanism specifically tailored for unbounded UDs.
2. COIN is claimed as the first defense mechanism effective against unbounded Unlearnable Datasets (UDs). Utilizing random pixel-based image transformations, COIN stands as an addition to the existing arsenal of defense strategies aimed at combating UDs.

**Weaknesses:**

1. The paper mentions the efficiency of the COIN method but falls short of providing a detailed computational analysis. Understanding the computational overhead is important for assessing the practicality of the method.

2. While the paper makes a significant contribution to image-based tasks, it does not explore the applicability of the COIN method to other model architectures and other types of data. This limitation narrows the paper's impact and leaves questions about its generalizability.

**Questions:**

1. What is the computational overhead for the intra-class matrix inconsistency and inter-class matrix consistency, compared with the computation cost of the baselines?
2. What's the performance of the proposed COIN on other tasks or other datasets?

---

> ### Author Response · Authors · 2023-11-16
> **Response to Reviewer R3xq (Part1):**
>
> > Q1. Provide a detailed computational analysis.
>
> A1. Thank you very much for your suggestion!  Assuming that the time complexity of each line of code in Algorithm 2 is ***O*(1)**, then for an unbounded UD $D_u=${$x_i \in R^{C\times H\times W}, i=1,2,...,N$}, the overall time complexity of performing COIN is ***O*(N×C×H×W)+*O*(N×H×W)+*O*(N)**.  Due to the fact that the values of C, H, and W of image $x_i$ are constant, *e.g.,* C=3, H=32, and W=32 for CIFAR-10 images, the final time complexity is optimized to: ***O*(N)+*O*(N)+*O*(N)=*O*(N)**. We appreciate your valuable feedback and have clarified this point in our revised paper. Your comments help us improve the clarity and quality of our research.
>
> Additionally, we used the same training parameter settings as mentioned in the main text to obtain the time overhead for each defense scheme on ResNet18 (Using one Nvidia GeForce 3090 GPU and Intel(R) Xeon(R) Gold 5218R CPU @ 2.10GHz), as shown in the table below, which demonstrates the efficiency of our scheme.
>
>
> |        Defenses         |  AT  | AVATAR |  AA  |  OP  | ISS-G | ISS-J | **COIN (Ours)** |
> | :---------------------: | :--: | :----: | :--: | :--: | :---: | :---: | :-------------: |
> | Time overhead (minutes) | 131  |  129   |  66  |  35  |  24   |  25   |       32        |
>
>
>
>
>
> > Q2. Transfer COIN to other architectures, other types of data, other datasets, and other tasks.
>
> A2. Thank you for your detailed review and valuable comments on our paper. The issue you raised regarding the applicability of the COIN method to different model architectures, data types, datasets, and tasks is indeed crucial. Allow me to elaborate on how we have addressed this concern.
>
>  Firstly, addressing different computer vision architectures, we have extended the application of the COIN method to two state-of-the-art architectures: GoogleNet [1] and InceptionNetV3 [2], using the CIFAR-10 dataset against CUDA. The experimental results, detailed in the table below, demonstrate a significant improvement in test accuracy (%), robustly attesting to the effectiveness of the COIN method.
>
> | Architectures↓Defenses→ |  w/o  |  MU   |  CM   |  CO   | DP-SGD |  AT   | AVATAR |  AA   |  OP   | ISS-G | ISS-J | COIN (Ours) |
> | :---------------------: | :---: | :---: | :---: | :---: | :----: | :---: | :----: | :---: | :---: | :---: | :---: | :---------: |
> |     InceptionNetV3      | 18.73 | 26.04 | 20.11 | 18.87 | 17.65  | 50.62 | 30.40  | 38.58 | 26.52 | 15.41 | 38.49 |  **72.88**  |
> |        GoogleNet        | 21.10 | 24.99 | 24.25 | 26.06 | 21.18  | 47.66 | 24.68  | 39.01 | 23.94 | 22.63 | 41.49 |  **69.07**  |
>
> Secondly, to further validate the adaptability of COIN, we have conducted experiments on a larger image dataset, ImageNet100, with an image size of 224×224. These additional experimental results have also been incorporated into our updated paper and are presented in the following table:
>
>
> | Defenses↓Models→ | ResNet18 | ResNet50 | DenseNet121 | MobileNetV2 |   **AVG**  |
> |:----------------:|:--------:|:--------:|:-----------:|:-----------:|:------:|
> |        w/o       |  25.74   |  26.66   |    21.70    |    16.30    | 22.60  |
> |        MU        |  34.96   |  19.38   |    27.78    |    15.60    | 24.43  |
> |        CM        |  16.54   |  24.04   |    23.58    |    8.00     | 18.04  |
> |        CO        |  25.46   |  29.20   |    23.90    |    17.58    | 24.04  |
> |        AT        |  **37.82**   | 36.80   |    30.34    |    41.42    | 36.60 |
> |       ISS-G      |  14.92   |  13.50   |    9.78     |    5.78     | 11.00  |
> |       ISS-J      |  30.10   |  **37.04**   |    25.52    |    28.04    | 30.18  |
> |    **COIN (Ours)**   |  37.80   |  35.38   |    **35.22**    |    **41.50**    | **37.48** |
>
>
>
>
> Regarding the applicability of the COIN method to other types of data and tasks, we note that the concept of Unlearnable Dataset (UD) currently predominates in the image domain for classification tasks, aiming to protect the privacy of image data. The feasibility and practical significance of UD in other data types or tasks have not been explored in the literature to date. Therefore, we are currently unable to extend the COIN defense method for UD to other data types or tasks. Our paper also refrains from making any claims regarding the effectiveness of COIN for other data types.
>
> Finally, I would like to express my gratitude once again for your insightful feedback. Your comments significantly contribute to enhancing the clarity and quality of our research. We also look forward to exploring the potential application of COIN to different data types and tasks in future studies, further expanding the scope of our research.
> We appreciate your attention and look forward to making further improvements to our paper.
>
>  [References]
>
> [1] [CVPR'15] Going Deeper with Convolutions
>
> [2] [CVPR'16] Rethinking the Inception Architecture for Computer Vision

---

> ### Author Response · Authors · 2023-11-16
> **Response to Reviewer R3xq (Part2):**
>
> > Q3. The computational overhead for the intra-class matrix inconsistency and inter-class matrix consistency, compared with the computation cost of the baselines?
>
> A3.  Thanks for your comments. The computational overheads of running calculations for intra-class matrix inconsistency and inter-class matrix consistency are both within one second on Intel(R) Xeon(R) Gold 5218R CPU @ 2.10GHz, with specific hyperparameter settings provided in Appendix A.2. Additionally, since we are the first to introduce the metrics of intra-class matrix inconsistency and inter-class matrix consistency specifically for unbounded UDs, there are currently no baselines available for comparison in calculating these two metrics.

---

> ### Author Response · Authors · 2023-11-20
> **Looking forward to further discussion**
>
> Dear Reviewer R3xq,
>
> We are grateful for your valuable comments. In our response, we have included the time complexity analysis for COIN. Moreover, we have tested COIN across more diverse architectures and larger datasets. We have incorporated all the changes in our revised manuscript for your kind consideration. And we hope that your concerns have been addressed.
>
> As you may be aware, we are approaching the final recommendation deadline for the reviewers. During this crucial period, we would appreciate the opportunity to engage in discussions with you. We are open to providing additional information based on your feedback or addressing any further questions you may have.
>
> If you find our response satisfactory, kindly consider updating your score. Please don't hesitate to reach out if you require any further clarification.
>
> Paper7861 Authors

---

> ### Author Response · Authors · 2023-11-23
> **We are still looking forward to further discussion**
>
> Dear Reviewer R3xq,
>
> With only two hours left until the final discussion and other reviewers actively participating in the discussion, we sincerely look forward to your positive response! If our replies address your concerns, we hope you kindly consider raising your score!
>
> Best wishes！
>
> Paper7861 Authors

---

### Official Review · Reviewer_ASN6 · 2023-11-01

**Soundness:** 3 good
**Presentation:** 3 good
**Contribution:** 3 good
**Rating:** 6
**Confidence:** 4

**Summary:**

Since recent defense methods are proven to be unsuccessful for resisting unbounded unlearnable examples, authors in this paper specifically design a defense called COIN to eliminate this threat. In detail, the authors first express unbounded UDs as a matrix-multiplication problem and propose the intra-class inconsistency and inter-class consistency matter based on empirical observation. Furthermore, they propose to employ randomly multiplicative transformation via interpolation operation as their method to defend unbounded unlearnable examples. The experiments demonstrate that COIN can achieve SOTA performance against on both small and large dataset across multiple architectures.

**Strengths:**

1 The technical novelty of this paper is good.

2 The writing of this paper is easy to follow.

3 The experiments are relatively convincing and comprehensive.

**Weaknesses:**

1 I think the observation about intra-class matrix inconsistency and inter-class matrix consistency is not closely correlated with the property of unbounded unlearnable attack. From my perspective, it essentially has no difference with the findings widely discussed in previous works: linearity separability is a important factor for crafting powerful unlearnable examples. Increasing intra-class matrix inconsistency or inter-class matrix consistency actually impair this separability to defend unlearnable examples.

2 Though the authors claim performing experiments on ImageNet, they only select a very small fraction of vanilla dataset (20 class, 2%) and resize the image to $64\times64$ which is still a low-resolution and small dataset. To further demonstrate the effectiveness of COIN on large dataset and considering the computational cost during pre-processing, I would recommend performing experiments on ImageNet-100 with $224\times224$ image size to further demonstrate the superiority of the proposed method.

3 From the perspective of integrity, I would suggest testing the performance of COIN  against OPS on CIFAR-100 dataset . Furthermore, combing Table 1 and Table 2 together to a large table will enhance the visual representation of experimental results for easy comparison.

4 It seems that the performance of COIN  is worse than those of ISS-J against OPS attack (-5%). However, COIN outperforms ISS-J a lot against CUDA. Why is that? Can you give more explanation?

**Questions:**

1 Acknowledging that COIN is a defense methods designed for unbounded UDs, however, in real situation, defenders do not have any knowlege about the kind of attacks adopted by attackers. Thus, I am curious whether COIN is effctive to defend bounded UDs.

2 Is COIN still effective when the perturbation for OPS is sample-wise [1]?

For other question, please refer to the weakness section.

[1] What Can We Learn from Unlearnable Datasets?

---

> ### Author Response · Authors · 2023-11-16
> **Response to Reviewer ASN6 (Part1):**
>
> > Q1. The intra-class matrix inconsistency and inter-class matrix consistency is not closely correlated with the property of unbounded unlearnable attack. Linearity separability is an important factor for crafting powerful unlearnable examples.
>
>
> Thank you for your insightful perspective! We appreciate the opportunity to clarify the distinction between our findings and the broader concept of linear separability in unlearnable attacks. It's important to note that linear separability, as discussed in the works of Yu *et al.* [1], primarily applies to additive noise in bounded UDs. However, not all additive noise exhibits linear separability, as indicated in [2]. This suggests that the principles governing bounded UDs may not directly apply to unbounded UDs.
>
> In our study, we specifically address the unique characteristics of unbounded UDs, which exhibit distinct noise patterns compared to bounded UDs. For example, as shown in Fig. 10, unbounded perturbations within the same class demonstrate non-identical noise in CUDA. This pattern of noise differs fundamentally from the class-wise noise in bounded UDs and necessitates a different analytical approach.
>
> Therefore, we introduced two new metrics—namely, intra-class matrix inconsistency and inter-class matrix consistency. These metrics are designed to quantify the unique properties of multiplicative matrix noise in unbounded UDs, which, in our observations, significantly influence their linear separability. By focusing on these metrics, we aim to provide a more nuanced understanding of how unbounded UDs behave and how they can be effectively countered.
>
> Our revised paper includes a detailed explanation of the logical relationship between these metrics and the linear separability of unbounded UDs. We believe this clarification will enhance the comprehensibility of our approach. Again, we thank you for your valuable feedback, which has greatly contributed to the improvement of our research.
>
> [References]
>
> [1] [KDD'22] Availability Attacks Create Shortcuts
>
> [2] [NeurIPS'23] What Can We Learn from Unlearnable Datasets?
>
>
>
>
>
>
> > Q2. Recommendation for ImageNet-100 with size 224x224 experiments.
>
> A2. Thank you for your valuable suggestion to conduct experiments on a larger and higher resolution dataset. In response to your recommendation, we have expanded our experimental setup to include ImageNet100 with 224x224 image size. This adjustment not only aligns with your suggestion but also allows us to test the effectiveness of COIN in a more challenging and realistic scenario.
>
> In our updated paper, we have replaced the initial experimental data from ImageNet20 (64x64) with these new results. These experiments on ImageNet100 not only demonstrate the adaptability of COIN to larger and higher resolution datasets but also reinforce the superiority of our approach.  We also show the experimental results as below:
>
> | Defenses↓Models→ | ResNet18 | ResNet50 | DenseNet121 | MobileNetV2 |   **AVG**  |
> |:----------------:|:--------:|:--------:|:-----------:|:-----------:|:------:|
> |        w/o       |  25.74   |  26.66   |    21.70    |    16.30    | 22.60  |
> |        MU        |  34.96   |  19.38   |    27.78    |    15.60    | 24.43  |
> |        CM        |  16.54   |  24.04   |    23.58    |    8.00     | 18.04  |
> |        CO        |  25.46   |  29.20   |    23.90    |    17.58    | 24.04  |
> |        AT        |  **37.82**   |  36.80    |    30.34    |    41.42    |  36.60 |
> |       ISS-G      |  14.92   |  13.50   |    9.78     |    5.78     | 11.00  |
> |       ISS-J      |  30.10   |  **37.04**   |    25.52    |    28.04    | 30.18  |
> |    **COIN (Ours)**   |  37.80   |  35.38   |    **35.22**    |    **41.50**    | **37.48**  |
>
> We believe these additional experiments substantially strengthen the evidence for COIN's effectiveness and are grateful for your feedback, which has been instrumental in enhancing the quality of our research.

---

> ### Author Response · Authors · 2023-11-16
> **Response to Reviewer ASN6 (Part2):**
>
> > Q3. Suggestion for testing COIN against OPS on CIFAR-100 and combining tables.
>
> A3. Thank you for your constructive suggestion to evaluate COIN's performance against OPS on the CIFAR-100 dataset. In response, we have incorporated additional experiments with OPS on CIFAR-100 into Tab. 1. These experiments were conducted following the same rigorous standards as our other tests, ensuring consistency and reliability in the results.
>
> Furthermore, based on your advice, we have combined Tab. 1 and Tab. 2 into a single, comprehensive table. This integration enhances the visual representation of our experimental results, facilitating easier comparison and better understanding of COIN's performance across different scenarios.
>
> The revised table showcases the superiority of COIN, especially in terms of average test accuracy (%) against OPS and CUDA, further supporting our claims about its effectiveness.
>
> We truly value and appreciate your feedback, which has significantly contributed to enhancing the clarity and quality of our research. The detailed experimental results and comparisons are now more effectively presented, thanks to your insightful comments.
>
>
>
>
> > Q4. Query on COIN's performance variance against OPS and CUDA UDs.
>
>
>
> A4. Thank you for your insightful query regarding the differential performance of COIN against OPS and CUDA UDs. The variation can be attributed to the distinct mechanisms of COIN's operation and the nature of the attacks it counters.
>
> In the case of OPS UDs, COIN's strategy of modifying global pixels, while effective against CUDA, inadvertently affects some essential image features. This is detailed in Sec. 5.4 of our paper. However, as shown in Fig. 7, by adjusting the parameter $\alpha$ in COIN, we can reduce its impact on global pixels, thereby enhancing its effectiveness against OPS. The selection of
> $\alpha$ is crucial; it is fine-tuned to strike a balance between defending against both CUDA and OPS with maximal efficacy.
>
> Conversely, ISS-J's performance lags behind COIN when countering CUDA UDs. This is primarily due to the fact that the extensive global multiplicative noise introduced by CUDA significantly degrades image features post JPEG's lossy compression, as illustrated in Fig. 10 (Appendix A.6). The lossy compression inherent to ISS-J exacerbates this feature loss, making it less effective against CUDA compared to COIN.
>
> We have expanded upon these explanations in our revised paper, providing a more comprehensive understanding of COIN's operational nuances and its adaptability to different types of unbounded UDs. This elucidation should help clarify why COIN outperforms ISS-J significantly against CUDA but shows a marginal lag against OPS.
>
>
>
>
>
> > Q5. The effectiveness of COIN against bounded UDs.
>
> A5. Thank you for your inquiry. We provided specific details in Tab. 10 in Appendix A.5.1, demonstrating that COIN has a defensive effect against certain bounded UDs.
>
>
>
>
> > Q6. The effectiveness of COIN against sample-wise OPS, *i.e.*, error minimizing noise + OPS.
>
> A6. COIN is ineffective against error-minimizing noise + OPS, as it is also ineffective against error-minimizing noise belonging to bounded noise. Thank you for your attention.

---

> ### Author Response · Authors · 2023-11-20
> **Looking forward to further discussion**
>
> Dear Reviewer ASN6,
>
> We are grateful for your valuable comments. In our response, for a more comprehensive study, we have included additional experiments on the COIN against CIFAR-100 OPS and ImageNet100 with image size of 224*224. Moreover, we have further clarified  differences between bounded and unbounded UDs in our revised paper. We have incorporated all the changes in our revised manuscript for your kind consideration. And we hope that your concerns have been addressed.
>
> As you may be aware, we are approaching the final recommendation deadline for the reviewers. During this crucial period, we would appreciate the opportunity to engage in discussions with you. We are open to providing additional information based on your feedback or addressing any further questions you may have.
>
> If you find our response satisfactory, kindly consider updating your score. Please don't hesitate to reach out if you require any further clarification.
>
> Paper7861 Authors

---

> > ### Comment · Reviewer_ASN6 · 2023-11-22
> > **Thank you for your reply.**
> >
> > Dear authors,
> >
> > I have carefully read your rebuttals. The performance of COIN on ImageNet100 dataset is still too marginal for me. However, in the favor of your efforts on rebuttals. I still decide to maintain my score unchanged. This is my final decision.
> >
> > Best,
> >
> > Reviewer ASN6

---

> > > ### Author Response · Authors · 2023-11-22
> > > **Thanks for your response!**
> > >
> > > Dear Reviewer ASN6，
> > >
> > >   We express our sincere gratitude for your comprehensive review and invaluable feedback throughout this process. Your insightful suggestions have significantly enriched the clarity and quality of our research. Thank you once again!
> > >
> > > Best,
> > >
> > > Paper7861 Authors

---

### Author Response · Authors · 2023-11-16
**General Response to all Reviewers:**

Thanks for your valuable comments and your appreciation of our **technical novelty** (**Reviewers ASN6, R3xq**), **good writing** (**Reviewers ASN6, T4ea**), **timely scheme** (**Reviewers R3xq, T4ea**), and **extensively convincing experiments** (**Reviewers ASN6, T4ea**).

We have revised and updated our paper based on your valuable suggestions. Here is the summary of updates that we've made to the draft:

* Added an explanation about the distinction in linear separability between additive noise in bounded UDs and multiplicative noise in unbounded UDs **(Reviewers ASN6, T4ea)**.

- Replaced the results obtained on ImageNet20 (64x64) with the results obtained on ImageNet100 (224x224) **(Reviewer ASN6)**.

- Added experiments on OPS using CIFAR-100 **(Reviewer ASN6)**.

- Added an explanation regarding the differences in the defense effectiveness of COIN and ISS-J against CUDA and OPS **(Reviewer ASN6)**.

- Added an analysis of the time complexity and the corresponding experimental results **(Reviewer R3xq)**.

- Adjusted the claim regarding the working scope of existing defenses and our finding about existing defenses **(Reviewer T4ea)**.

---

### Meta-Review · Area_Chair_nJjc · 2023-12-08

**Metareview:**

While the paper presents a novel approach to defending against unbounded UDs, significant concerns remain about its practical applicability, the relevance of its theoretical underpinnings, and its performance compared to other defense mechanisms like ISS-J. The reviewers suggest that additional experiments and a more comprehensive demonstration of COIN's effectiveness across various scenarios and datasets would be necessary to substantiate the claims made in the paper. The current limitations, particularly in terms of generalizability and theoretical justification, suggest that the paper may not yet meet the standards for acceptance.

**Justification For Why Not Higher Score:**

As said in meta review

**Justification For Why Not Lower Score:**

NA

---

### Decision · Program_Chairs · 2024-01-16

Reject